# Endosomal LC3C-pathway selectively targets plasma membrane cargo for autophagic degradation

Paula P. Coelho [1,2], Geoffrey G. Hesketh[3,9], Annika Pedersen [1,4,9], Elena Kuzmin[1,2], Anne-Marie N. Fortier [1], Emily S. Bell[5], Colin D. H. Ratcliffe [1,2], Anne-Claude Gingras [3,6] & Morag Park [1,2,7,8 ✉]

Autophagy selectively targets cargo for degradation, yet mechanistic understanding remains incomplete. The ATG8-family plays key roles in autophagic cargo recruitment. Here by mapping the proximal interactome of ATG8-paralogs, LC3B and LC3C, we uncover a LC3C-Endocytic-Associated-Pathway (LEAP) that selectively recruits plasma-membrane (PM) cargo to autophagosomes. We show that LC3C localizes to peripheral endosomes and engages proteins that traffic between PM, endosomes and autophagosomes, including the SNARE-VAMP3 and ATG9, a transmembrane protein essential for autophagy. We establish that endocytic LC3C binds cargo internalized from the PM, including the Met receptor tyrosine kinase and transferrin receptor, and is necessary for their recruitment into ATG9 vesicles targeted to sites of autophagosome initiation. Structure-function analysis identified that LC3C-endocytic localization and engagement with PM-cargo requires the extended carboxy-tail unique to LC3C, the TBK1 kinase, and TBK1-phosphosites on LC3C. These findings identify LEAP as an unexpected LC3C-dependent pathway, providing new understanding of selective coupling of PM signalling with autophagic degradation.

[1] Rosalind and Morris Goodman Cancer Institute, McGill University, Montreal, QC, Canada. [2] Department of Biochemistry, McGill University, Montreal, QC H3G 1Y6, Canada. [3] Lunenfeld-Tanenbaum Research Institute, Mount Sinai Hospital, 992A-600 University Avenue, Toronto M5G 1X5, Canada. [4] Department of Experimental Medicine, McGill University, Montreal, Canada. [5] Department of Biochemistry and Molecular Biology, Pennsylvania State University, State College, PA 16802, USA. [6] Department of Molecular Genetics, University of Toronto, Toronto M5S 1A8, Canada. [7] Department of Medicine, McGill University, Montreal, Canada. [8] Department of Oncology, McGill University, Montreal H2W 1S6, Canada. [9] These authors contributed equally: Geoffrey G. Hesketh, Annika Pedersen. ✉email: morag.park@mcgill.ca

Macroautophagy (hereafter referred to as autophagy) is a catabolic process that leads to the disassembly of macromolecules and recycling of their constituents to be reused by the cell[1]. It is a mechanism for maintaining cellular homeostasis that is present in most cells at a basal level and is enhanced in response to cellular stressors[2]. Autophagy is marked by de novo formation of a double-membrane structure called a phagophore, which expands to engulf cytoplasmic content before closing to form a mature autophagosome that fuses with lysosomes where cargo degradation occurs[2].

Although autophagy was originally thought to be a process for bulk degradation, it is increasingly understood that cargo can be selectively recruited into autophagosomes, including proteins, organelles, and pathogens[3,4]. Selective autophagy is emerging as a means to modulate intracellular signaling[5–7] and plasma membrane (PM) receptors[8–11] via targeting of signaling proteins into autophagic pathways in a process termed signalphagy[12,13]. Given the important role of many signaling molecules in modulating key biological processes, understanding how autophagy regulates signaling networks and what dictates specificity in this process is important in health and disease.

Many transmembrane receptors are canonically regulated via intracellular trafficking. Receptors internalized from the PM into the endocytic network can either be recycled back to the PM or targeted towards late endosomes and subsequent lysosomal degradation, the balance of which determines signaling output and biological responses[14]. The autophagic pathway is now recognized to be tightly interconnected with the endocytic trafficking network, and early and late recycling endosomes have been found to donate membrane to nascent phagophores[15]. However, how the interconnection between trafficking endocytic vesicles and autophagic membranes contributes to the selective targeting of PM-derived cargo to autophagosomes is not yet mechanistically understood.

The ATG8-family of proteins play crucial roles at multiple stages of the autophagic pathway. These are small cytoplasmic proteins that become conjugated to the membrane of nascent phagophores and play a key role in cargo recruitment[16]. Although present as a single protein in yeast, in humans this family has expanded to include six orthologs, divided into two subfamilies: (i) LC3A, LC3B, and LC3C, and (ii) GABARAP, GABARAP-L1, and GABARAP-L2/GATE-16[16]. This expansion in higher eukaryotes is thought to provide increased flexibility and selectivity to autophagic responses[17]. Among the LC3-family, LC3B is best studied and most widely used marker for autophagosomes[18], whereas LC3C has the highest sequence divergence, with an extended carboxy-terminal tail (C-tail), the role of which remains unclear. LC3C has been linked to selective events within the autophagic pathway[19–22], including redirecting PM-derived Met receptor tyrosine kinase (RTK)[11] from endosomal compartments to the autophagic-degradative pathway, however, the mechanism for LC3C selectivity in this process is unknown.

Here we have mapped proximal interactomes of the ATG8 members, LC3B and LC3C, and uncovered a previously undescribed LC3C-dependent selective autophagy pathway for PM-derived cargo. We show that a subset of cellular LC3C localizes to peripheral endosomes in a process that is regulated by the serine–threonine kinase, TBK1, and requires the LC3C extended C-tail. Endosomal LC3C engages with PM-derived cargo, including Met-RTK and transferrin receptor 1(TfR), and functions to recruit these proteins for autophagic degradation. We provide a mechanistic understanding of how LC3C trafficking and autophagic pathways interconnect, demonstrating how autophagy can modulate cellular signaling via LC3C-mediated selective degradation of PM-derived cargo.

## Results

**LC3B and LC3C engage distinct proximal interactomes.** To understand selectivity within the autophagic process and elucidate how LC3C differs from LC3B in engaging cargo and modulating biological responses, we performed BioID in HeLa cells, where we established a selective role for LC3C in the degradation of the Met RTK[11], under basal conditions. The abortive biotin ligase (BirA*) was fused to bait proteins, LC3C or LC3B, allowing for biotinylation of proximal proteins[23], and bait expression was verified by western blotting (Supplementary Fig. 1a). Using the SAINTexpress computational tool[24] to identify high-confidence interactions (defined as FDR < 1%), we identified 264 and 182 proximal interactors for LC3B and LC3C respectively (Supplementary Data 1). Although LC3C and LC3B are associated with a large set of common proteins, each displayed distinct proximal interactors (Fig. 1a, b), supporting both overlapping and non-redundant functions for LC3C and LC3B. To validate by an orthogonal approach, we performed co-immunoprecipitation of co-expressed tagged baits and hits. For this, we selected hits appearing selective for LC3C, LC3B or with equal abundance for both baits. Of eight hits first tested, all validated to have the predicted selectivity, highlighting the robustness of the BioID approach (Fig.1c, d).

Core-autophagy proteins were identified as shared proximal interactors between LC3C and LC3B. These include components of the machinery responsible for conjugating LC3-proteins to the membrane of nascent phagophores, such as ATG7 and ATG3[25], as well as known cargo receptors that simultaneously bind different cargo and ATG8-proteins (p62/SQSTM1, NBR1, CCPG1, RETREG1/FAM134B)[26] (Fig. 1e). Other cargo receptors demonstrated specificity for LC3C over LC3B, including OPTN and the previously identified NDP52[21] (Fig. 1e). Unexpectedly, we identified selective proximal interactions of LC3C, but not LC3B, with the first components of the autophagy machinery recruited to pre-autophagosomal structures (PAS). This includes the interaction of LC3C with ATG13 and FIP200, members of the initiation complex[25,27], as well as ATG9, a multipass transmembrane protein known to traffic in vesicles providing membranes to growing phagophores[28–30] (Fig. 1e). Since ATG13, FIP200 and ATG9 are transiently recruited to phagophores and are not present in mature autophagosomes[27,30], this suggests that LC3C may be recruited to PAS earlier and through a separate pathway from LC3B.

To investigate what underlies these differential proximal interactions and their functional consequence, we analyzed the predicted localization of all hits identified on the screen using both a manually curated list of subcellular protein localization (Fig.1f, Supplementary Data 1) and the SubCell Barcode database[31] (Supplementary Fig. 1b). Both approaches revealed that LC3C-specific proximal interactors were significantly enriched for proteins localizing to the PM and endosomes when compared with LC3B-specific proximal interactors (Fig. 1g and Supplementary Fig. 1c), supporting that LC3C itself may have a higher propensity for localization to these subcellular compartments.

**LC3C engages with proteins connecting PM, endosomes, and autophagosomes.** Given the distinct profiles of the proximal interactomes identified for LC3B and LC3C, we performed indirect immunofluorescence, under basal conditions used in the BioID screen, in addition to starvation-induced autophagy, to determine the localization of LC3C and LC3B. Quantification of the overall distribution profile revealed distinct localizations for LC3C and LC3B under both conditions (Fig. 2a, b, Supplementary Fig. 2a, b). Overall, LC3C displayed enhanced localization to puncta within peripheral and intermediate cellular compartments

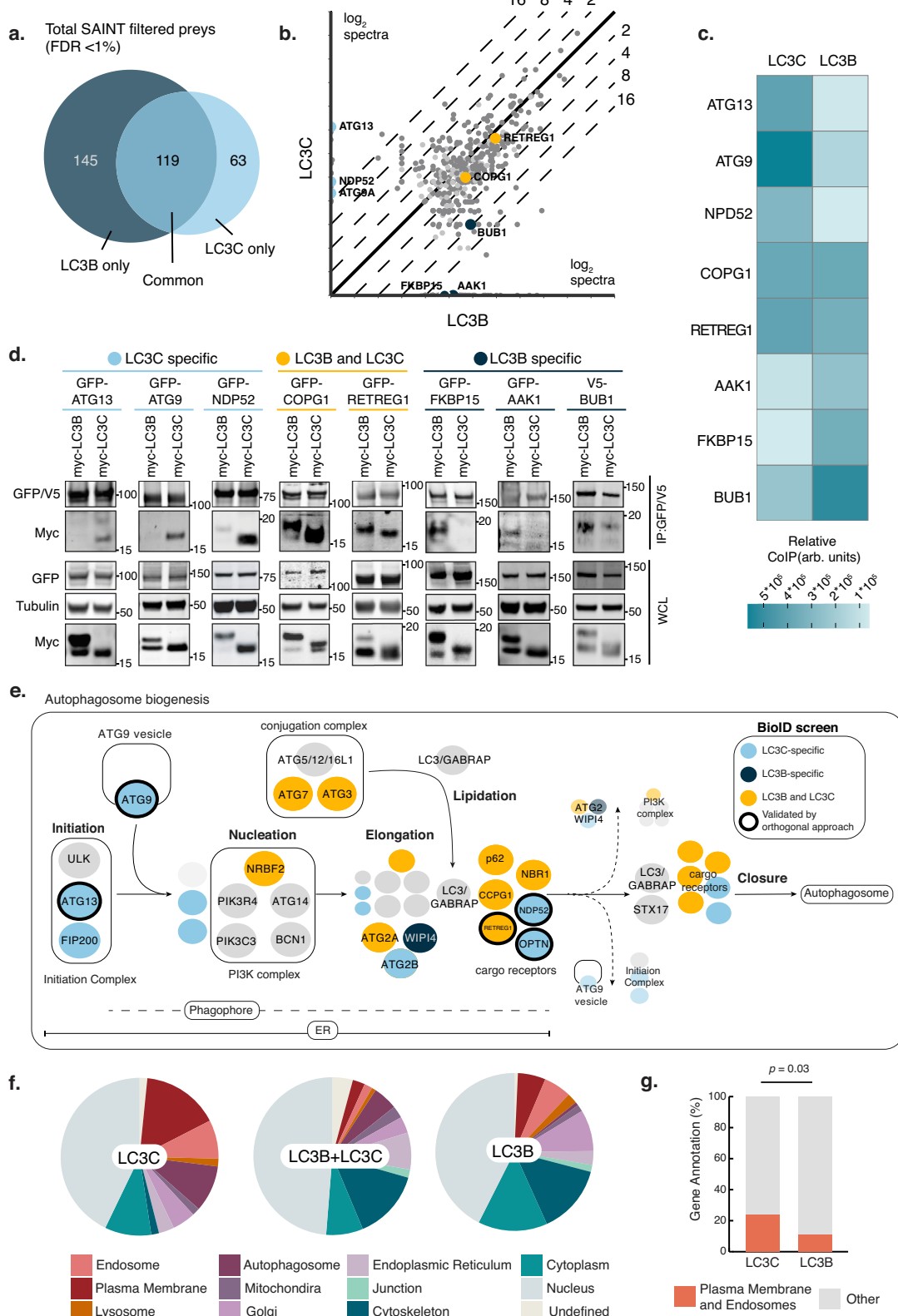

when compared to LC3B, which localized predominantly to puncta in a perinuclear compartment, a typical position for mature autophagosomes[32]. Differential localization of LC3C to the cell periphery was enhanced following starvation (Fig. 2a, b), with a three-fold increase in the proportion of LC3C at the peripheral compartment when compared to basal conditions (4% vs 13%), and as such subsequent experiments were performed

under starvation conditions. Consistent with this distinct localization, many LC3C- but not LC3B-specific proximal interactors predicted to localize to endosomes, displayed secondary annotations to the PM or autophagosomes (Fig. 2c, Supplementary Data 1). This indicated that LC3C may interact with proteins transitioning through these compartments. In support of this, co-immunoprecipitation of LC3C, but not LC3B, was observed with

**Fig. 1 LC3B and LC3C engage distinct proximal interactomes. a** Venn diagram of proximity interactions with False Discovery Rate (FDR) <1% for LC3C and LC3B baits (see Supplementary Data 1 for complete list). **b** Bait versus bait plot of abundance (average spectral counts plotted as log2 values) for all proximity interactions (dark gray = FDR ≤ 1%, light gray = 1% < FDR ≤ 5%) identified throughout the 24 h-labeling period. Proximity interactions validated by an orthogonal approach are color coded in light-blue for LC3C-specific, dark-blue for LC3B-specific and yellow for shared (see Fig. 1c, d). The black diagonal line indicates equal abundance for both baits, with relative fold enrichment indicated by dashed lines. **c** Heat-map displaying quantification of co-immunoprecipitation (CoIP) experiments in **d**. Values indicate the amount of myc-LC3 detected in the immunoprecipitate (IP) normalized by an amount in the whole-cell lysate (WCL) and standardized using WCL tubulin loading control as arbitrary units (arb. units) (N = 3 for each bait). Source data are provided as a Source Data file. **d** Western blots of co-immunoprecipitation experiments of GFP or V5- tagged proximal interactors used as baits for pulldown of myc-LC3B and/or myc-LC3C demonstrating preferential or shared interaction with the LC3-constructs. Prior to lysis, cells were starved for 2 h in HBSS. **e** Schematic demonstrating sequential recruitment of core autophagy machinery during autophagosome biogenesis with emphasis on key complexes and proteins identified in LC3-BioID screen. Light-blue represents LC3C-specific hits, dark blue represents LC3B-specific, and yellow represents shared proximal interactors. Black circles indicated hits validated by an orthogonal approach (See also Figs. 1c, d, 2e and Supplementary Fig. 2c). Other proteins involved in the specified processes are included for completeness and represented in gray. **f** Subcellular location distribution of LC3 proximal-interactors based on a combination of gene ontology (GO) cellular compartment annotation and GeneCards cards data. **g** Two-sided Fisher's exact test comparing the abundance of hits localized to PM and endosomes in the LC3C and LC3B proximal interactomes from (**f**).

hits predicted to localize to endosomes (RUFY1), to PM and endosomes (TfR, Caveolin-1/CAV1, and VAMP3), and to endosomes and autophagosomes (ATG9, OPTN, and TBK1) (Fig. 2d, e and Supplementary Fig. 2c). Collectively, the association of LC3C with these proteins supports a functional role for LC3C within an endocytic compartment connecting PM to autophagosomes.

To investigate if the endocytic localized LC3C was required to recruit endocytosed cargo towards autophagic degradation, we further examined LC3C proximal interactors that link endosomes to autophagosomes. Notably, ATG9-positive vesicles are known to localize to peripheral puncta[33] and endosomes in response to starvation-induced stress[34], which is an essential step for autophagosome formation. Moreover, the SNARE vesicular transport protein VAMP3 localizes to ATG9-positive endocytic vesicles[35], is associated with fusion events between endocytic and autophagic structures[35,36], and is implicated in the recycling of PM proteins[37,38]. Thus, both VAMP3 and ATG9 are identified as crucial components of a pathway linking endosomes to autophagosomes and both selectively interact with LC3C over LC3B by forward and reverse coimmunoprecipitations (Supplementary Fig. 2d). The association of VAMP3 and ATG9 with LC3C positive compartments was further confirmed by immunofluorescence, where VAMP3 and ATG9 display enhanced colocalization with LC3C puncta, (34% and 45% respectively), when compared to LC3B puncta (21% and 19% respectively) (Fig. 2f, g). Together these data provide support for a preferential role for LC3C over LC3B in an endocytic compartment connected to the autophagic pathway.

**LC3C localizes with PM-derived cargo in vesicles marked by VAMP3 and ATG9.** To establish if LC3C preferentially engages with PM-derived cargo within endosomes, we examined the Met-RTK, which is targeted for autophagic degradation dependent on LC3C[11,39]. In response to stimulation by its ligand, hepatocyte-growth-factor (HGF), Met internalizes into endosomes via clathrin-mediated endocytosis (CME)[40,41]. Cells treated with dynasore, a dynamin inhibitor that blocks CME[42], decreased Met-LC3C co-immunoprecipitation, supporting that Met internalization is required for Met-LC3C interaction (Supplementary Fig. 3a).

Following ligand stimulation, Met internalizes to Rab5-positive early-endosomes, from which it either traffics towards lysosomal degradation or recycles back to the PM via Rab4-positive recycling endosomes[43], a process which regulates Met down-stream signaling[44] and cell migration[45–47]. Met/LC3C puncta display significant overlap with Rab5 and Rab4 (65% and 66% overlap respectively, Supplementary Fig. 3b) as well as with

VAMP3 and ATG9 (66% and 28% overlap respectively; Fig. 3a–d), demonstrating that Met/LC3C localize together with multiple endocytic as well as autophagic markers. To determine if Met and LC3C interactions were dependent on VAMP3 or ATG9, we performed knockdown (KD) of each. This revealed a requirement for VAMP3 but not ATG9 for Met-LC3C co-immunoprecipitation (Fig. 3e, f). This supports a sequential model whereby VAMP3 is necessary to deliver Met to an endocytic compartment where interaction with LC3C occurs, followed then by engagement of a Met-LC3C complex with ATG9. Consistent with this we observe that Met/LC3C and Met/LC3C/VAMP3-positive puncta display similar peripheral distributions, whereas Met/LC3C/ATG9 positive puncta are enriched towards intermediate and perinuclear compartments consistent with LC3C and Met joining ATG9-positive vesicles at later stages (Fig. 3g).

**LC3C recruits cargo to ATG9 vesicles and nascent autophagosomes.** Since ATG9-positive vesicles are recruited to PAS, and donate membranes to growing phagophores[48,49], we investigated if LC3C was required for Met recruitment into ATG9-vesicles targeted to autophagosome initiation sites. Using cells with CRISPR-Cas9-mediated LC3C deletion (LC3Cdel)[11], we observed a decrease in colocalization of Met and ATG9 upon loss of LC3C (Fig. 4a, b). This was accompanied by retention of Met in vesicles at the cell periphery in LC3Cdel cells following HGF stimulation (Fig. 4c) and increased cell-surface levels of Met (Fig. 4d, Supplementary Fig. 4e, g), consistent with decreased trafficking of Met to a degradative autophagic compartment upon LC3C loss and enhanced recycling.

To investigate if LC3C-vesicles and their Met-RTK cargo are delivered to nascent phagophores we employed live-cell imaging, tracking a Met-HGF complex using fluorescently labeled HGF (HGF-555), together with GFP-tagged markers of PAS, ATG13 and FIP200, identified in the LC3C-proximal interactome (Fig. 1e). This revealed that both GFP-ATG13 and GFP-FIP200 engage with HGF-555-positive puncta over time (Fig. 4e and Supplementary Fig. 4c), with 42% of ATG13 and 51% of FIP200 puncta becoming positive for HGF-555/Met throughout the 10 min imaging period, placing Met in a molecularly defined phagocytic compartment. Next, we assessed whether LC3C was required for Met RTK recruitment to PAS. We established that LC3C-deletion did not affect the frequency or duration of ATG13 or FIP200-positive puncta (Supplementary Fig. 4a, b). In contrast, the overlap of ATG13 and FIP200-puncta with HGF-555 in LC3Cdel cells was significantly decreased to 14.7% and 19.8%, respectively, when compared to control cells (42% and 51%; Fig. 4f, Supplementary Fig. 4d) and displayed shorter duration, from an average of 28 s and 31 s to 16 s and 17 s, respectively

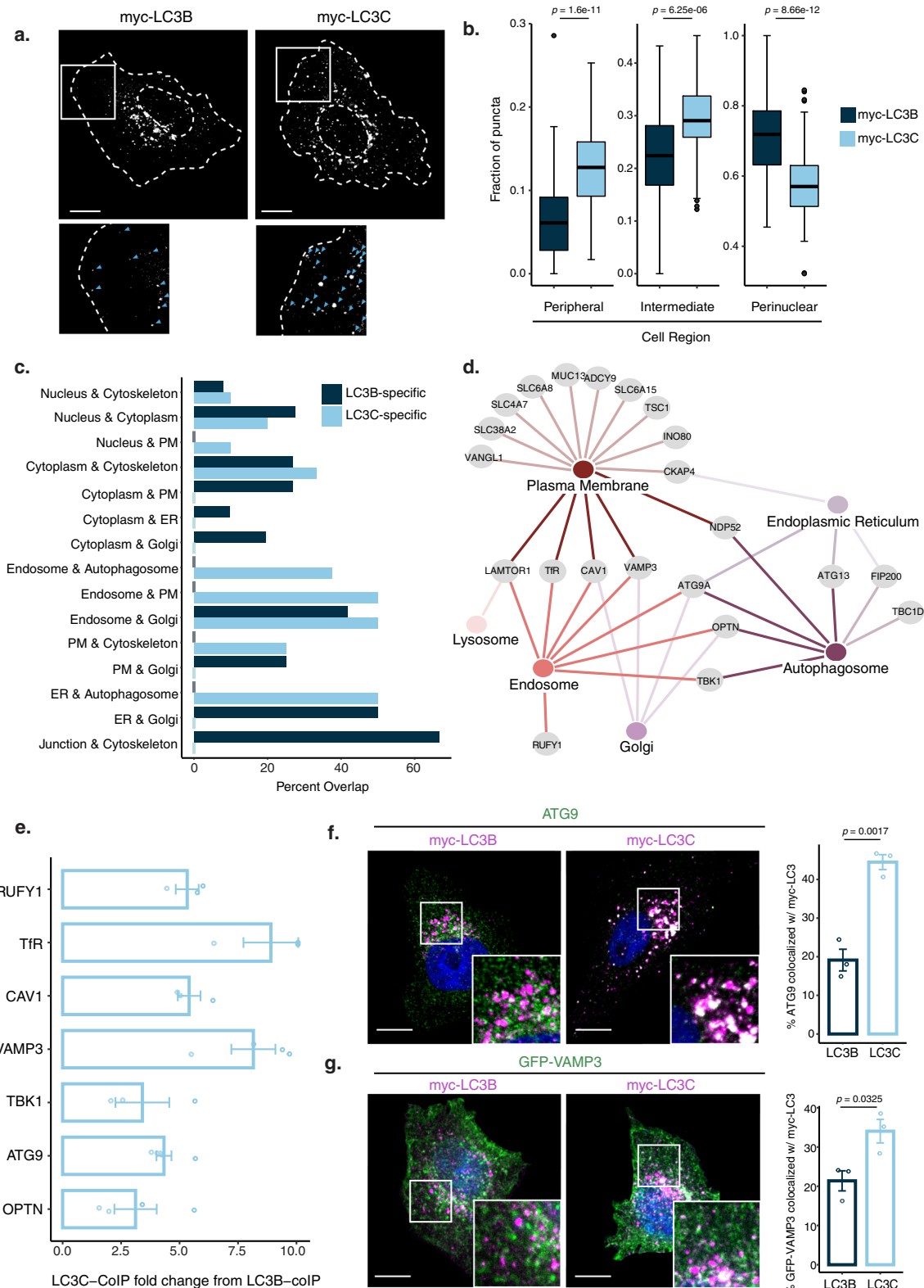

(Fig. 4g, Supplementary Fig. 4d). Given the transient nature of ATG13 and FIP200 recruitment to PAS[27,30], this indicates that Met is present at the early stages of autophagosome biogenesis and supports that an LC3C-dependent pathway can deliver PM-derived cargo, such as Met, to nascent phagophores.

We postulated that other PM-receptors that undergo endocytosis and recycling could be similarly recruited by LC3C to ATG9

vesicles. From the LC3C BioID data, the transferrin receptor (TfR) emerged as a likely candidate (Fig. 2d). In contrast to the Met RTK, TfR is constitutively internalized and recycled without ligand-stimulation[50], making it readily identifiable in the baseline BioID screen conducted here. TfR is tightly regulated via intracellular trafficking[51], is present in ATG9 vesicles[30,52], and colocalizes with phagophore markers[53] and VAMP3[38]. Following

**Fig. 2 LC3C engages with proteins connecting PM, endosomes, and autophagosome. a** Representative images of intracellular distribution of myc-LC3B and myc-LC3C transfected into HeLa cells following 2 h starvation in HBSS. At the bottom, blue arrowheads indicate myc-puncta. Scale bar = 10 μm. **b** Quantification of myc-LC3B and myc-LC3C distribution; fraction of puncta identified across three subcellular cellular regions (peripheral, intermediate and perinuclear) are plotted. Box plots indicate median (middle line), 25th, 75th percentile (box), and 1.5 IQR (Interquartile range) of the nearer quartile (whiskers) and outliers (single points). 60–80 cells were quantified per condition across three experiments (unpaired two-sample *t*-test). **c** Secondary subcellular locations were annotated for the LC3B and LC3C proximal interactomes based on a combination of GO cellular compartment annotation and GeneCards cards data. For each compartment with a minimum of three hits, the percentage of interactors with secondary annotations to other subcellular locations is plotted. **d** Interaction network of LC3C-specific interactors. Hubs represent subcellular compartments, while nodes represent hits identified in the BioID screen. Edges connect proteins to all compartments it has annotated associations with. **e** Quantification of co-immunoprecipitation (CoIP) experiments of predicted LC3C-specific interactors annotated to localize at intersections of PM, endosomes, and autophagosomes. Graph displays fold change increase in pulldown of myc-LC3C compared to myc-LC3B using GFP- or V5-tagged predicted interactors as bait. (*N* = 3 for RUFY1, TfR, CAV1, TBK1, *N* = 4 for OPTN, VAMP3 and *N* = 5 for ATG9, mean ± SEM). See also Supplementary Fig. 2c. **f** Representative images of HeLa cells transfected with myc-LC3C or myc-LC3B fixed following 2 h starvation in HBSS and stained for myc-tag and endogenous ATG9. Colocalization was quantified from three independent experiments (mean ± SEM, unpaired two-sample *t*-test, Scale bar = 10 μm). **g** Representative images of HeLa cells transfected with myc-LC3C or myc-LC3B and GFP-VAMP3 fixed following 2 h starvation in HBSS and stained for myc-tag. Colocalization was quantified from three independent experiments (mean ± SEM, unpaired two-sample *t*-test, Scale bar = 10 μm). Source data for **b**, **e**, **f**, **g** are provided as a Source Data file.

confirmation that TfR-LC3C coimmunoprecipitate as predicted from the screen (Supplementary Fig. 2c), we investigated if LC3C was necessary for effective autophagic targeting of TfR by examining TfR trafficking in LC3Cdel cells. Loss of LC3C led to decreased colocalization of TfR with ATG9-puncta, from 32% in control to 22% in LC3Cdel cells (Fig. 4h, i). Notably, this was accompanied by a redistribution of TfR-positive vesicles towards the periphery, with a decrease in TfR-puncta in the perinuclear region (Fig. 4j) and increased cell-surface levels of TfR (Fig. 4k, Supplementary Fig. 4f) in LC3Cdel cells, consistent with a requirement for LC3C in TfR targeting to a perinuclear autophagic-degradative compartment. Collectively, these analyses of the LC3C-interactome have allowed the identification of an unexpected trafficking route, whereby LC3C is recruited to VAMP3-positive endosomes, where it can engage with internalized PM-cargo, such as the Met RTK or TfR, and is necessary for subsequent recruitment of these cargos into ATG9 vesicles targeted to PAS.

**LC3C localization to peripheral endosomes requires the LC3C C-tail.** To investigate what underlies the selective localization of LC3C, but not LC3B, to a peripheral endocytic compartment linked to the ATG9-network, we employed a structure-function approach. Although all ATG8 family members are highly conserved, LC3C uniquely possesses an extended C-terminal tail (Fig. 5a). By generating an LC3C mutant with C-tail deletion (ΔCtail) we established that the LC3C-C-tail was necessary for the localization of LC3C to peripheral puncta (Fig. 5b, c). Consistent with this, deletion of the C-tail led to the loss of interaction of LC3C with cargo, including Met and TfR, as well as with BioID hits annotated to localize to endosomes, RUFY1 and TBK1 (Fig. 5d, e). Together these data support that the extended C-tail of LC3C promotes localization to peripheral endosomes enhancing LC3C interaction with proteins with endosomal localization: Met, TfR, TBK1, and RUFY1.

All LC3-proteins undergo a proteolytic cleavage of the C-terminal peptide to be recruited onto autophagosomes. This process is catalyzed by the cysteine protease ATG4[25], and serves to expose a glycine residue that is subsequently lipidated to the membrane of autophagosomes[25]. Functions for the LC3 C-tail prior to cleavage are still poorly defined. Recent data supports that, Arg134 on the extended LC3C C-tail can form an intramolecular salt bridge when LC3C is phosphorylated on Ser93 and Ser96[54] (Supplementary Fig. 5a). This interaction can sequester the C-tail, decreasing access to ATG4 and subsequent cleavage, therefore decreasing LC3C lipidation[54]. Such delayed lipidation could in turn delay the recruitment of LC3C to mature

autophagosomes and enhance LC3C retention in other compartments such as endosomes. To test whether phosphorylation at Ser93/96 regulates LC3C distribution, we generated an LC3C mutant lacking these phosphosites (LC3C-S93/96A) (Supplementary Fig. 5b). Consistent with our hypothesis, this mutant displayed reduced localization to puncta at the cell periphery when compared to LC3C-WT, and displayed an enriched localization to puncta in a perinuclear region, characteristic of mature autophagosomes (Fig. 5f, g).

To confirm a role for the LC3C C-tail in regulating LC3C-cargo we assayed the ability of each mutant to rescue LC3C deletion phenotypes. We have previously established[11] that LC3C deletion leads to a delayed degradation of the Met RTK following HGF stimulation under starvation conditions (Supplementary Fig. 5c). Overexpression of LC3C-WT but not LC3C ΔCtail or S93/96A mutants rescues LC3C-dependent Met degradation (Supplementary Fig. 5c). In a similar manner, elevated levels of steady-state TfR were observed in LC3Cdel cells when compared to controls, these were reversed by overexpression of LC3C-WT but not LC3C ΔCtail or S93/96A mutants (Supplementary Fig. 5d). Together these data support that both the LC3C C-terminal tail and S93/96 phosphosites, are necessary for LC3C-dependent degradation of Met and TfR.

**LC3C engages cargo prior to its recruitment to autophagosomes.** To establish whether engagement of LC3C with Met or TfR requires other components of the autophagy machinery, we knocked-down members of different complexes involved in autophagosome biogenesis and assessed how this impacted LC3C-localization and protein interactions. We found that LC3C retains the ability to co-immunoprecipitate with Met and TfR following KD of ATG3, ATG5, and ATG7, components of the autophagy conjugation system responsible for lipidating LC3-proteins to autophagosomes following cleavage of the C-tail (Fig. 6a, b). Hence, LC3C lipidation is not essential for its engagement with Met and TfR. Similarly, KD of members of the autophagy inhiation and nucleation complexes, ATG13 and ATG14 respectively, did not abrogate co-immunoprecipitation of LC3C with Met or TfR (Supplementary Fig. 6a, b). Supporting that LC3C engages with Met and TfR prior to autophagosome formation. Moreover, although KD of ATG7 and ATG14 increased cytoplasmic localization of LC3C, consistent with a decreased conjugation to autophagic membranes, vesicle-bound LC3C-puncta were observed throughout these cells upon permeabilization to remove unbound protein (Supplementary Fig. 6c, Supplementary Movie 1). Importantly, the remaining LC3C positive puncta in ATG7 or ATG14 KD cells, colocalized

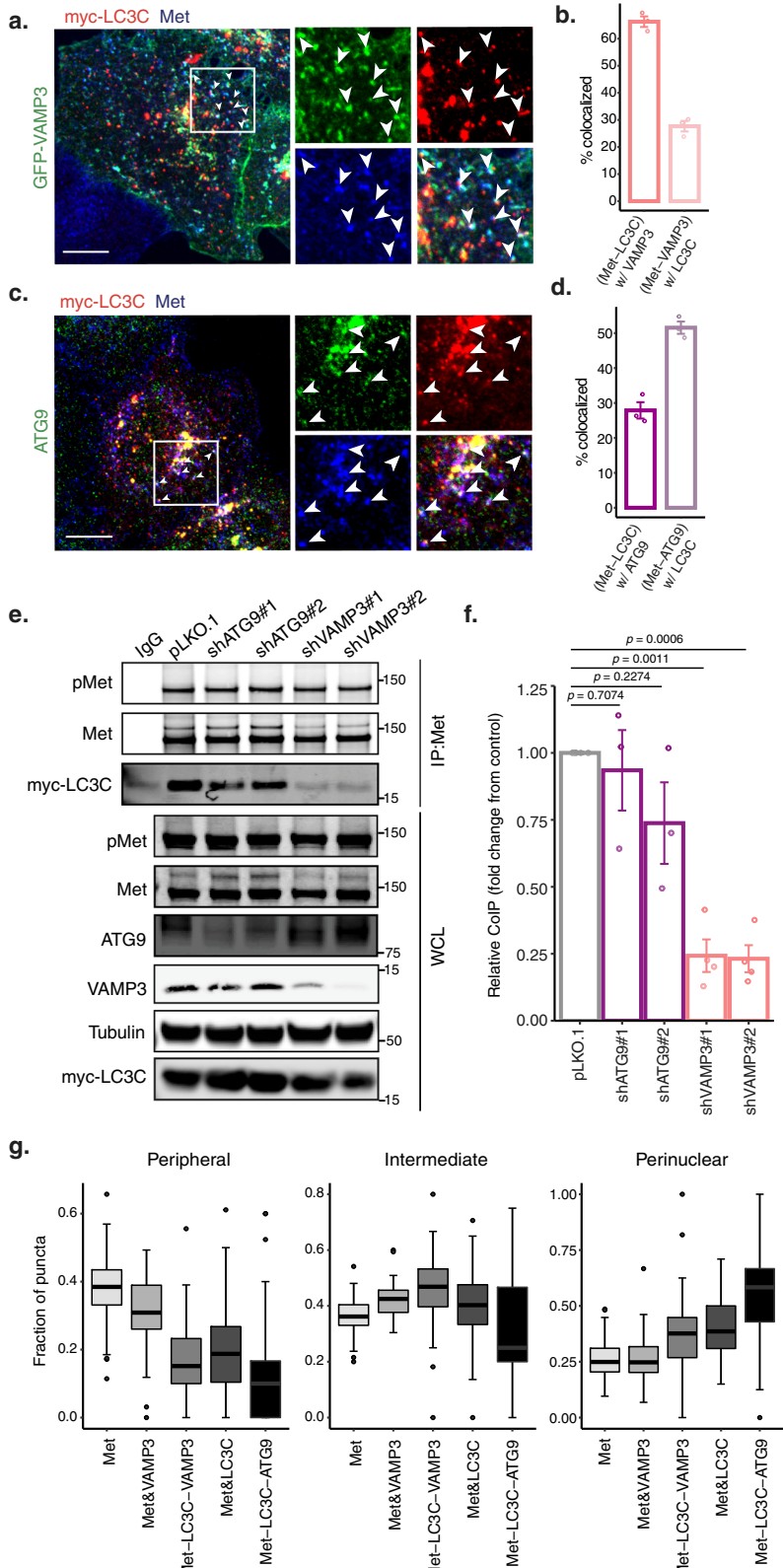

with Met and endocytic markers, Rab4 and Rab5, to similar levels as control cells (60%, 61% and 61% overlap with Rab5, and 56%, 48%, and 53% overlap with Rab4 for ATG7-KD, ATG14-KD and controls respectively, Fig. 6c, d, Supplementary Fig. 6d, e). We previously established that KD of essential autophagy genes including ATG5, ATG7, and ATG14 lead to stabilization of Met, decreasing Met RTK degradation, and

increasing Met-dependent signaling[11]. Taken together, these results support that LC3C acts upstream from these core-autophagy proteins and that while LC3C can still interact with Met in the absence of such proteins, disruption of normal autophagosome biogenesis abrogates autophagic degradation of Met. Overall, this identifies a functional role for LC3C prior to its conjugation to autophagosomes, whereby unlipidated LC3C

**Fig. 3 LC3C localizes with PM-derived cargo in vesicles marked by VAMP3 and ATG9. a** HeLa cells were transfected with myc-LC3C and GFP-VAMP3, starved, and stimulated with HGF for 15 min to trigger Met-internalization prior to fixation. Arrowheads indicate triple colocalization of endogenous Met, myc-LC3C, and GFP-VAMP3. Scale bar = 10 μm. **b** Quantification of colocalization of immunofluorescence experiments shown in **a** (values represent mean ± SEM, $N = 3$). **c** HeLa cells were transfected with myc-LC3C, starved, and stimulated with HGF for 15 min to trigger Met-internalization prior to fixation. Arrowheads indicate triple colocalization of endogenous Met, myc-LC3C, and ATG9. Scale bar = 10 μm. **d** Quantification of colocalization of immunofluorescence experiments shown in **c** (values represent mean ± SEM, $N = 3$). **e** Co-immunoprecipitation (CoIP) of endogenous Met and myc-LC3C following knockdown (KD) of either ATG9 or VAMP3 with two different shRNA guides each. Cells were transfected, starved, and stimulated with HGF for 20 min prior to lysis. **f** Quantification of western blot CoIP in **e**. Graph displays fold change from pLKO.1-empty vector control ($N = 4$, mean ± SEM, one sample *t*-test from a hypothetical value of 1.0). **g** Quantification of the distribution of single, dual or triple colocalized puncta on images from **a** and **c** across three subcellular regions (peripheral, intermediate, and perinuclear). Box plots indicate median (middle line), 25th, 75th percentile (box), and 1.5 IQR (Interquartile range) of the nearer quartile (whiskers) and outliers (single points). 55–75 cells were quantified per condition across three experiments. Source data for graphs in **b**, **d**, **f**, **g** are provided as a Source Data file.

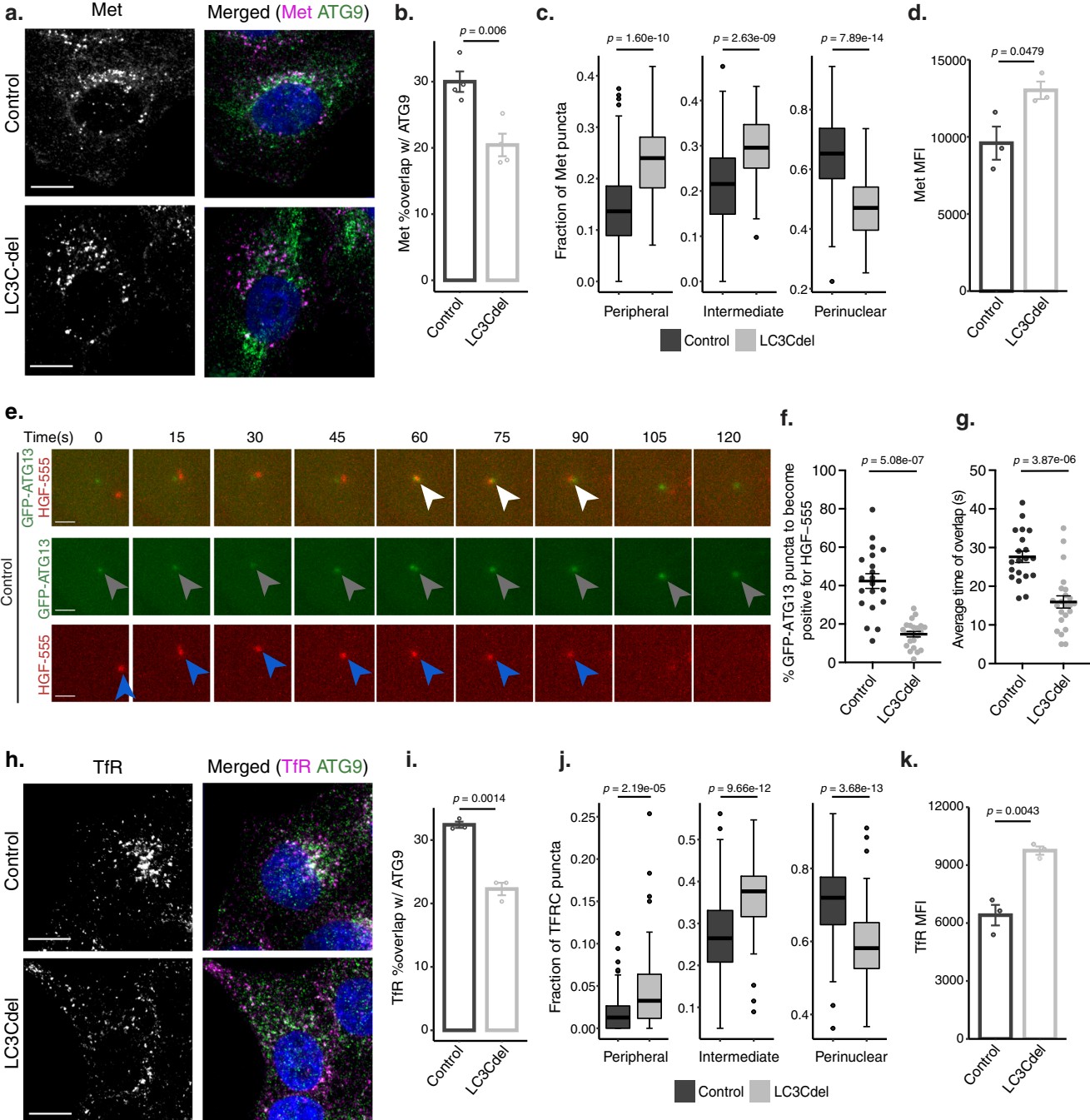

**Fig. 4 LC3C recruits cargo to ATG9 vesicles and nascent autophagosomes. a** Representative images of control and LC3Cdel cells fixed and stained for endogenous Met and ATG9 following starvation in HBSS and 30min-HGF-stimulation to induce Met-trafficking. Scale bar =10 µm. **b** Quantification of colocalization of Met and ATG9 as shown in **a** ($N = 4$, mean ± SEM, unpaired two-sample $t$-test). **c** Quantification of fraction of Met-puncta across three subcellular regions (peripheral, intermediate, and perinuclear) in control and LC3Cdel cells. Box plots indicate median, 25th, 75th percentile (box), and 1.5 IQR (Interquartile range) of the nearer quartile (whiskers) with outliers as points. 70-100 cells were quantified per condition across four experiments (unpaired two-sample $t$-test). **d** Mean Fluorescent Intensity (MFI) of cell surface Met detected by flow cytometry following HBSS starvation and 30 min-HGF-stimulation ($N = 3$, mean ± SEM, unpaired two-sample $t$-test). **e** Representative images of control cells expressing GFP-ATG13 starved in HBSS prior to stimulation with HGF-555. Depict Met/HGF-555 (blue arrowhead) trafficking to GFP-ATG13-puncta (gray arrowhead). White arrowhead mark frames with colocalization. Scale bar = 1.5 µm. **f** Quantification of the proportion of GFP-ATG13 puncta to become positive for HGF-555. Twenty cells were quantified for Control and 23 for LC3Cdel across two experiments (mean ± SEM, unpaired two-sample $t$-test). **g** Quantification of average duration of overlap between GFP-ATG13 and HGF-555. 20 cells were quantified for Control and 23 for LC3Cdel across two experiments (mean ± SEM, unpaired two-sample $t$-test). **h** Representative images of control and LC3Cdel cells starved in HBSS and stained for endogenous TfR and ATG9. Scale bar = 10 µm. **i** Quantification of colocalization of TfR and ATG9 as shown in **h** ($N = 3$, mean ± SEM, unpaired two-sample $t$-test). **j** Quantification of the fraction of TfR puncta across three subcellular regions (peripheral, intermediate, and perinuclear) in control and LC3Cdel cells. Box plots indicate median, 25th, 75th percentile (box), and 1.5 IQR of the nearer quartile (whiskers) with outliers as points. 90–100 cells were quantified per condition across three experiments (unpaired two-sample $t$-test). **k** MFI of cell surface of TfR detected by flow cytometry following HBSS starvation ($N = 3$, mean ± SEM, unpaired two-sample $t$-test). Source data for all graphs are provided as a Source Data file.

can act to recruit endocytic cargo, Met, and TfR towards nascent autophagosomes.

**TBK1 regulates LC3C retention in peripheral endosomes**. The serine–threonine kinase TBK1 can directly phosphorylate LC3C at Ser93/96[54], and was identified in our BioID screen as an LC3C-specific proximal interactor, which was validated via co-immunoprecipitation (Fig. 2d, Supplementary Fig. 2c). TBK1 has diverse functions linked to its subcellular localization[55]. TBK1 is found in autophagic[56] and endocytic[57] membranes and has been linked to ATG9 vesicles[55]. To establish if TBK1 is a potential regulator of LC3C we examined TBK1 localization. We established that TBK1 localizes with LC3C in endocytic compartments, with TBK1/LC3C-positive puncta colocalizing with Rab5 and Rab4 (31% and 57% respectively, Fig. 7a, Supplementary Fig. 7b). To investigate if TBK1 regulates LC3C localization, cells were treated with a TBK1 small molecule inhibitor (TBK1i)-MRT67307 (Supplementary Fig. 7c). This led to redistribution of LC3C towards a perinuclear region (Fig. 7b, c), similar to that observed with the LC3C-S93/96A-mutant lacking TBK1 phosphorylation sites. This is consistent with enhanced localization of LC3C to mature autophagosomes in the absence of TBK1 activity and TBK1 phosphorylation on S93/96. In support of reduced retention of LC3C to an endocytic compartment following TBK1-inhibiton, LC3C displayed decreased co-immunoprecipitation with endosomal cargo, Met and TfR under these conditions (Fig. 7d), and both receptors displayed reduced ability to reach a perinuclear compartment, consistent with diminished recruitment into autophagosomes (Supplementary Fig. 7d, e). Together these data support that TBK1 contributes to LC3C phosphorylation, which in turn delays LC3C lipidation and allows for retention of LC3C on endosomal membranes where it engages cargo, Met and TfR.

## Discussion
Selective degradation of cellular components by autophagy is increasingly recognized as critical in shaping how cells respond to cellular stressors, stimuli, and disease states, highlighting a need for a thorough understanding of mechanisms of cargo recruitment for autophagic degradation. Here with the aid of an unbiased BioID screen, we identify and characterize an unexpected LC3C-mediated autophagy pathway that can selectively engage protein trafficking from the PM. We show that a population of LC3C is enriched to peripheral puncta and colocalizes with endosomal and recycling markers, Rab5 and Rab4, and the

snare protein VAMP3. Enrichment of LC3C in endocytic structures is dependent on the LC3C C-tail, TBK1, and TBK1 phosphosites on LC3C; and allows for engagement with internalized cargo, Met and TfR, and their subsequent recruitment into ATG9-vesicles targeted to sites of autophagosome initiation. We have termed this new trafficking route for LC3C-dependent autophagic degradation of endocytosed cargo as the LC3C-endocytic-associated-pathway (LEAP) (Fig. 7e).

In a canonical view of autophagy, LC3-proteins are incorporated into nascent autophagic membranes to recruit and engulf cargo, and subsequently, facilitate autophagosome-lysosome fusion for cargo degradation[32]. LEAP is one of several emerging atypical autophagy pathways that have begun to uncover an expanding complexity and diversity of LC3 protein functions[58]. Importantly we distinguish LEAP from the other recently characterized pathways known to link the trafficking network to autophagy, LC3B-associated phagocytosis (LAP)[59] or LC3B-associated endocytosis (LANDO)[60]. Both LAP and LANDO are independent of components of the initiation complex, including ATG13 and FIP200, and are not known to involve ATG9 vesicles, relying instead on direct LC3 lipidation at vesicular membranes[59,60]. In contrast, LEAP involves localization of unlipidated LC3C to endocytic compartments where it acts to recruit cargo, such as the Met RTK, to ATG9-positive vesicles targeted to sites of autophagosome initiation marked by ATG13 and FIP200. Cargo can then be incorporated into nascent autophagosomes for degradation. This LEAP model is consistent with our BioID data that identified distinct proximal interactomes for LC3C and LC3B, with an enrichment of proteins localized to the PM, endosomes, and the autophagy initiation complex among the LC3C-specific interactors.

An increased understanding of how atypical forms of autophagy can be differentially regulated is of particular importance. We reveal how LEAP can be selectively modulated dependent on the LC3C unique extended C-terminal tail. Despite cleavage of this peptide preceding LC3-lipidation onto autophagic membranes, we[11] and others[61] have described how the presence of the LC3C C-terminal peptide is necessary for LC3C-dependent selective cargo binding. We now provide a mechanistic understanding of how this may occur; showing that the C-terminal peptide is a key determinant of LC3C localization to an endocytic compartment and subsequent cargo engagement. Importantly, we establish that this is dependent on TBK1 and the presence of known TBK1 phosphosites at Ser93/96 on LC3C, which have been shown to contribute to a conformational change of the LC3C C-tail[54]. TBK1 has already been implicated in selective

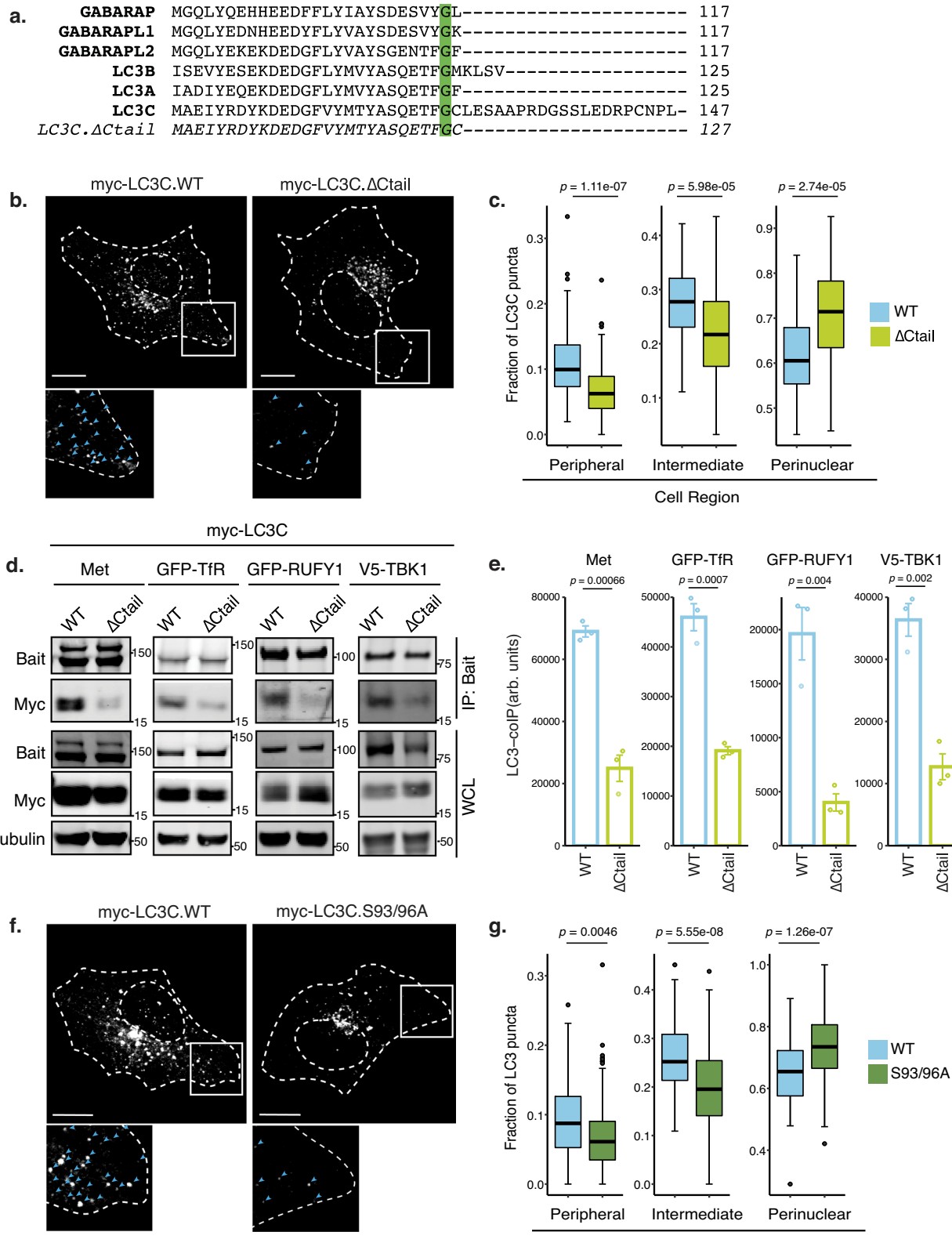

degradation of depolarized mitochondria (mitophagy)[62] and intracellular pathogens (xenophagy)[63]. Here we describe a new role for TBK1 in selective autophagy, via modulation of LEAP. Using a TBK1 small molecule inhibitor, we established that TBK1 not only regulates LC3C localisation to peripheral endosomal puncta, but also the ability of LC3C to bind PM-derived cargo, Met, and TfR. Hence this work adds new functional significance

to previous observations that phosphorylation of LC3C by TBK1 decreases ATG4-mediated LC3C cleavage and subsequent lipidation[54]. In our model, we show how TBK1 potentiates retention of unlipidated LC3C on endosomes and consequently facilitates cargo recruitment into LEAP (Fig. 7e).

The need for flexibility to fine-tune autophagic responses in higher eukaryotes is evidenced by the expansion of the ATG8-

**Fig. 5 LC3C localization to peripheral endosomes requires the LC3C C-tail. a** Schematic depicting the carboxy-tail of all ATG8 family members, as well as the LC3C-ΔCtail deletion mutant. **b** Representative images of the subcellular distribution of myc-LC3C-WT and myc-LC3C-ΔCtail. Transfected cells were starved for 2 h in HBSS and fixed prior to immunofluorescent staining for myc. At the bottom, blue arrowheads indicate myc-puncta. Scale bar = 10 μm. **c** Quantification of subcellular distribution of myc-LC3C from **b**. The fraction of puncta across three subcellular cellular regions (peripheral, intermediate, and perinuclear) was quantified per cell. Box plots indicate median (middle line), 25th, 75th percentile (box), and 1.5 IQR (Interquartile range) of the nearer quartile (whiskers) and outliers (single points). 65–85 cells were quantified per condition across four experiments (unpaired two-sample $t$-test). **d** Co-immunoprecipitation (CoIP) experiments of LC3C-WT and LC3C-ΔCtail with LC3C-interactors used as bait. Endogenous Met, overexpressed GFP-TfR, V5-TBK1 or GFP-RUFY1 were immunoprecipitated following lysis of starved cells. In the case of Met an additional 20 min HGF-stimulation was used to trigger Met-activation and internalization. **e** Quantification of CoIP experiments in **d**. Values indicate the amount of myc-LC3C detected in the immunoprecipitate (IP) normalized by an amount in the whole-cell lysate (WCL) and standardized using WCL tubulin loading control; values denote arbitrary units (arb. units) ($N = 3$ for each bait, mean ± SEM, unpaired two-sample $t$-test). **f** Representative images of intracellular distribution of myc-LC3C-WT and myc-LC3C-S93-96A. Transfected cells were starved 2 h in HBSS prior to fixation and myc immunofluorescent staining. At the bottom, blue arrowheads indicate myc-puncta. Scale bar = 10 μm. See also Supplementary Fig. 5b. **g** Quantification of the subcellular distribution of myc-LC3C from **f**. The fraction of puncta across three subcellular cellular regions (peripheral, intermediate, and perinuclear) was quantified per cell. Box plots indicate median (middle line), 25th, 75th percentile (box), and 1.5 IQR (Interquartile range) of the nearer quartile (whiskers) and outliers (single points). 60–80 cells were quantified per condition across four experiments (unpaired two-sample $t$-test). Source data for graphs in **c**, **e**, **g** are provided as a Source Data file.

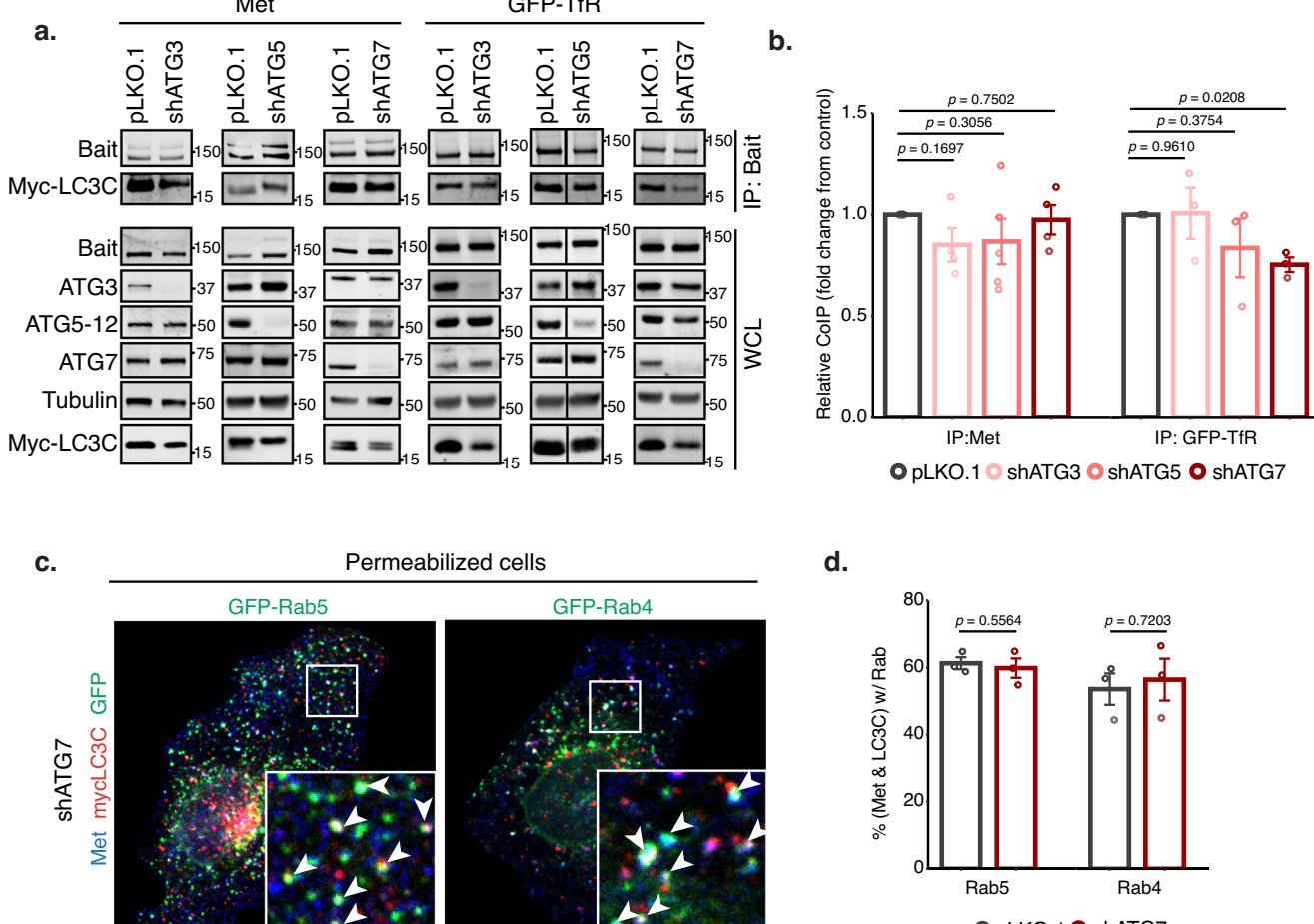

**Fig. 6 LC3C engages cargo prior to its recruitment to autophagosomes. a** Co-immunoprecipitation (CoIP) of endogenous Met or GFP-TRFC with myc-LC3C in HeLa cells stably expressing shATG3, shATG5, shATG7 or empty vector (pLKO.1). Cells were starved and, for the Met CoIPs, stimulated with HGF for 20 min prior to lysis. Show Met and TfR can still interact with LC3C following the knock-down of autophagic conjugation machinery. **b** Quantification of western blot CoIP in **a**. Graph displays fold change from pLKO.1-empty vector control ($N = 4$ for Met CoIPs, $N = 3$ for TfR CoIPs, mean ± SEM, one sample $t$-test from a hypothetical value of 1.0). **c** Representative image based on three experimental replicates of HeLa cells stably expressing shATG7 transfected with myc-LC3C and either early endosome marker, GFP-Rab5, or fast-recycling endosome marker, GFP-Rab4, starved and stimulated with HGF for 15 min to trigger Met-internalization prior to cell permeabilization and fixation. Shows Met-LC3C colocalize in endocytic structures following the loss of ATG7. Arrowheads indicate triple colocalized puncta. Scale bar = 10 μm. **d** Quantification of colocalization of immunofluorescence experiments performed in cells with stable expression of shATG7, from (**c**), compared to empty vector (pLKO.1) treated and fixed in the same manner ($N = 3$, values represent mean ± SEM, unpaired two-sample $t$-test). Source data for graphs in **b** and **d** are provided as a Source Data file.

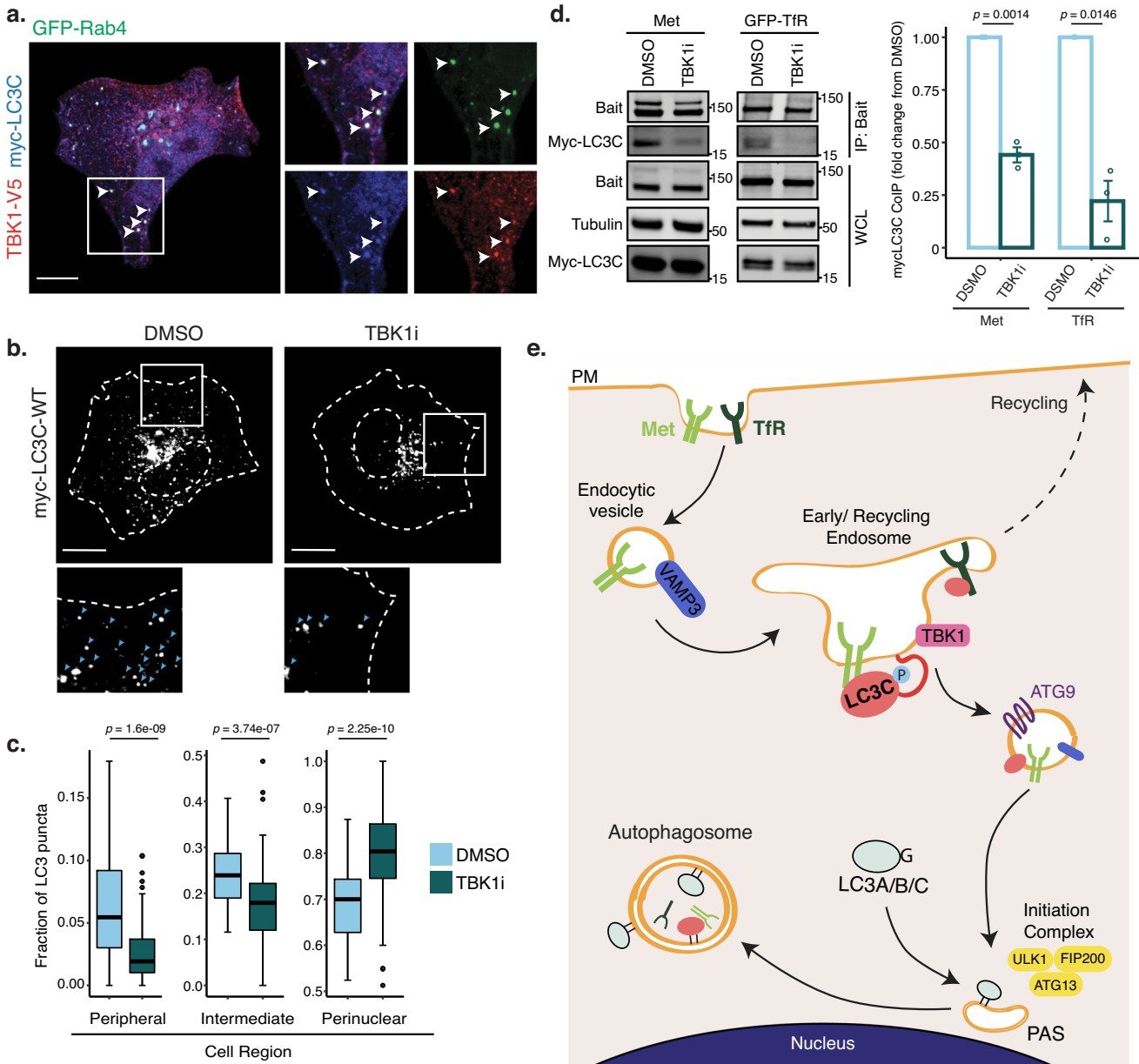

**Fig. 7 TBK1 regulates LC3C retention in peripheral endosomes. a** Representative images showing colocalization of V5-TBK1, GFP-Rab4, and myc-LC3C in transfected HeLa cells starved for 2 h in HBSS and fixed prior to immunofluorescent staining for myc and V5. Arrowheads indicate colocalization of all three proteins. Scale bar = 10 μm. **b** Representative images demonstrating the subcellular distribution of myc-tagged LC3C-WT in transfected HeLa cells starved for 2 h with TBK1 inhibition (TBK1i, MRT67307 at 10uM) or DMSO control. At the bottom, blue arrowheads indicate myc-puncta. Scale bar = 10 μm (See also Supplementary Fig 7c). **c** Quantification of the subcellular distribution of myc-LC3C upon treatment with TBK1 inhibitor or DMSO control as shown in **b**. Graphs show the fraction of puncta identified across three subcellular regions (peripheral, intermediate, and perinuclear). Box plots indicate median (middle line), 25th, 75th percentile (box), and 1.5 IQR (Interquartile range) of the nearer quartile (whiskers) and outliers (single points). 70–90 cells were quantified per condition across four experiments (unpaired two-sample t-test). Source data are provided as a Source Data file. **d** Co-immunoprecipitation (CoIP) experiments of myc-LC3C with either endogenous Met or overexpressed GFP-TfR, following a 2 h treatment with TBK1-inhibition (10 μM) or DMSO-control. For Met CoIP, 20 min HGF-stimulation was used to trigger Met-activation and internalization. The relative CoIP was calculated by measuring the amount of myc-LC3C detected in the immunoprecipitate (IP) normalized by the amount of myc-LC3C in the whole-cell lysate (WCL) and standardized using WCL tubulin loading control. The fold change from the untreated condition was plotted for each bait used ($N = 3$, mean ± SEM, one sample t-test from a hypothetical value of 1.0). Source data are provided as a Source Data file. **e** Model of LC3C-selective autophagy pathway. LC3C phosphorylation in endosomes by kinases such as TBK1 promotes a conformational change in the LC3C-C-tail, which delays recognition and processing by ATG4 and consequent rapid conjugation into autophagosomes. This delay and concomitant retention within endocytic structures allows LC3C to bind internalized cargo, such as Met or TfR. LC3C can then recruit this cargo into ATG9-vesicles targeted to nascent phagophores for cargo incorporation into autophagosomes, and eventual degradation.

family combined with a growing list of known autophagy cargo receptors, and an emerging understanding of the variability and specificity of LC3 protein recognition mechanisms and functions[26]. We demonstrate that LEAP provides a new yet unexpected mechanism whereby cells can integrate extracellular signals with the intracellular state of cells through context-dependent recruitment of cargo, such as PM-receptors. LC3C has been described to be selectively downregulated in many cancers[64,65] suggesting LEAP may be uncoupled from general autophagy in these tumors, potentially contributing to the elevation of Met[66] and TfR[67] observed in many human cancers. Further investigation of these parallel autophagy pathways may offer important clues on how to exploit autophagy therapeutically.

## Methods

**Cell culture and stable cell line generation.** For BioID, HeLa Flp-In T-REx cells (Thermo Fisher Scientific) were grown in complete growth media consisting of DMEM with 4.5 g/L glucose and 4 mM L-glutamine (Thermo Scientific, 11965) supplemented with 5% FBS, 5% Cosmic calf serum, 100 U/mL penicillin, 100 μg/mL streptomycin. The Flp-In and T-REx loci in HeLa cells were maintained by three passages in media containing 100 μg/mL zeocin and 15 μg/mL blasticidin prior to establishing stable lines. Tagged (BirA*-FLAG) constructs were generated by Gateway cloning from sequence-validated entry vectors generated by PCR from cDNA clones (MAP1LC3B (NM_022818.5) and MAP1LC3C (NM_001004343.3)) mutants were generated by PCR mutagenesis and sequence validated. Polyclonal populations of stable HeLa Flp-In T-REx cells were generated as described[23]. Integrated BirA*-FLAG-tagged genes were selected and maintained with 200 μg/mL hygromycin B, and expression was induced for 24 h by the addition of tetracycline at 1 μg/mL final concentration. Parental cell lines were negative for mycoplasma contamination (MycoAlert, Lonza).

Other experiments were performed using HeLa cells (ATCC) maintained in DMEM (Gibco) with 10% fetal bovine serum (Gibco) at 37 °C and 5%CO₂. For generation of stable ATG9, VAMP3, ATG3, ATG5, ATG7, and ATG14 knockdown cells, lentiviral shRNA vectors were retrieved from the arrayed Mission®TRC genome-wide shRNA collections purchased from Sigma-Aldrich Corporation. The following lentiviral shRNA vectors targeting ATG9 (#1: TRCN0000148385, #2: TRCN0000129286), VAMP3 (#1: TRCN0000029814, #2: TRCN0000029815), ATG3 (TRCN0000148120), ATG5(TRCN0000330392), ATG7 (TRCN0000007584), ATG14 (TRCN0000142647) and empty pLKO.1 vector was used. Viral particles were produced by co-expressing shRNA or control constructs with packaging plasmids psPAX2 and pMD2.G in HEK-293T (obtained from Frank Graham, McMaster University) cells using calcium phosphate transfection protocol. Media containing viral particles was collected and passed through a 0.45 μm filter. Cells were treated with virus in media containing 8 μg/ml polybrene. Forty-eight hours after transduction cells were selected and maintained in 2 μg/ml puromycin dihydrochloride (Sigma). Transient siRNA-mediated knockdown of ATG13 was performed using ON-TARGETplus ATG13 siRNA— SMARTpool from Dharmacon (L-020765-01-0005, GE Healthcare). LC3C knockout cell lines were generated using CRISPR/Cas9, and preciously described[11]. The sequence of all oligonucleotides used can be found in Supplementary Table 2.

**BioID.** Cells were grown to ~75% confluency and bait expression and biotin labeling was induced simultaneously (1 μg/ml tetracycline, 50 μM biotin). After 24 h, cells were rinsed and scraped into 1 mL of PBS. Cells were collected by centrifugation (500 × g for 3 min) and stored at −80 °C until further processing. Cell pellets were thawed on ice, and a 4:1 (v/w) ratio of ice-cold lysis buffer was added (50 mM Tris-HCl, pH 7.5, 150 mM NaCl, 1% Nonidet P-40 substitute (IGEPAL-630), 0.4% SDS, 1.5 mM MgCl₂, 1 mM EGTA, benzonase, protease inhibitors). Cells were resuspended with a P1000 pipette tip (~10–15 aspirations), and subjected to a rapid freeze/thaw cycle (dry ice to 37 °C water bath). Lysates were rotated at 4 °C for 30 min, then centrifuged at 16,000 × g for 20 min at 4 °C. Supernatants were collected and incubated with 20 μL (packed beads) of pre-washed streptavidin-Sepharose (GE) with rotation overnight at 4 °C. Beads were collected (500 × g for 2 min), the supernatant discarded, and the beads transferred to new tubes in 500 μL of lysis buffer. Beads were washed once with SDS wash buffer (50 mM Tris-HCl, pH 7.5, 2% SDS), 2× with lysis buffer, and 3× with 50 mM ammonium bicarbonate, pH 8.0 (ABC; all wash volumes are 500 μL with centrifugation at 500 × g for 30 sec between each wash). Beads were resuspended in 100 μL of ABC containing 1 μg of sequencing grade trypsin and gently mixed at 37 °C for 4 h. One microgram of fresh trypsin was added and the samples were allowed to digest overnight. The supernatant was collected (by centrifugation at 500 × g for 2 min) and the beads were washed with 100 μL of molecular biology grade H₂O and pooled with peptides. Digestion was terminated by acidification with formic acid (2% final concentration) and peptides were dried by vacuum centrifugation.

**Mass spectrometry acquisition using TripleTOF mass spectrometers.** Each sample (5 μL in 2 % formic acid; corresponding to 1/8th of a 15 cm tissue culture dish) was directly loaded at 800 nL/min onto an equilibrated HPLC column. The peptides were eluted from the column over a 90 min gradient generated by a Eksigent ekspert™ nanoLC 425 (Eksigent, Dublin CA) nano-pump and analyzed on a TripleTOF™ 6600 instrument (AB SCIEX, Concord, Ontario, Canada). The gradient was delivered at 400 nL/min starting from 2% acetonitrile with 0.1% formic acid to 35% acetonitrile with 0.1% formic acid over 90 min followed by a 15 min clean-up at 80% acetonitrile with 0.1% formic acid, and a 15 min equilibration period back to 2% acetonitrile with 0.1% formic acid, for a total of 120 min. To minimize carryover between each sample, the analytical column was washed for 2 h by running an alternating sawtooth gradient from 35% acetonitrile with 0.1% formic acid to 80% acetonitrile with 0.1% formic acid at a flow rate of 1500 nL/min, holding each gradient concentration for 5 min. Analytical column and instrument performance were verified after each sample by loading 30 fmol bovine serum albumin (BSA) tryptic peptide standard with 60 fmol α-casein tryptic digest and running a short 30 min gradient. TOF MS mass calibration was performed on BSA reference ions before running the next sample to adjust for mass drift and verify peak intensity. Samples were analyzed in data-dependent acquisition (DDA) mode. The DDA method consisted of one 250 milliseconds (ms) MS1 TOF survey scan from 400 to 1800 Da followed by ten 100 ms MS2 candidate ion scans from 100 to 1800 Da in high sensitivity mode. Only ions with a charge of 2+ to 5+ that exceeded a threshold of 300 cps were selected for MS2, and former precursors were excluded for 7 s after one occurrence.

**SAINT analysis and data visualization.** SAINTexpress (version 3.6.1 (45)) was used to score proximity interactions from MSPLIT-DIA data. SAINTexpress calculates, for each prey protein identified by a given bait, the probability of a true proximity interaction relative to negative control runs using spectral counting as a proxy for abundance. Bait runs (two biological replicates each) were compared against thirty-two negative control runs consisting of fourteen BirA*-FLAG-only samples, fourteen 3xFLAG-only samples, and four EGFP-BirA*-FLAG samples (compressed to twelve "virtual controls" that maximize stringency of scoring and mimic the worst-case scenario of a protein being detected because it is endogenously biotinylated in the absence of BirA* or frequently biotinylated by expression of recombinant BirA*). Preys with a false discovery rate (FDR) < 1% (Bayesian estimation based on the distribution of the Averaged SAINT scores across both biological replicates) were considered high-confidence proximity interactions. Bait-vs-bait (bait versus bait) plots were generated using ProHits-viz[68] (prohits-viz.org). In ProHits-viz, once a prey passes the selected FDR threshold (here 1%) with at least one bait, all its quantitative values across the dataset are retrieved for all baits.

**Subcellular localization analysis.** SubCellBarcode classification: Subcellular localization data was downloaded from the SubCellBarcode database (subcellbarcode.org). The 'Neighborhood Class' and 'Compartment Class' was then noted for each of the hits identified in the BioID screen. Data from four different cancer cell-lines (H322, HCC827, U251, A431) were analyzed separately to confirm trends observed were consistent and not cell-type specific.

Manual curation: To have a more extensive coverage of hits identified a manual curation process was also employed. For this all Gene Ontology (GO) term-name and GO-domain terms linked to the hits identified was downloaded from BioMart (from Esembl Gene 88; seast.ensembl.org/info/data/biomart/index.html). Based on analysis of the GO terms and the most prominent location associated to a given gene was noted. For genes with diverse locations and function, data from GeneCards (genecards.org) and accompanying classification from the COMPARTMENTS database (compartments.jensenlab.org) was used for final annotations (as per data available in 2018). The final classification for each gene is noted on Supplementary Data 1.

**Constructs and transfections.** All constructs used are noted in Supplementary Table 1. Novel LC3C mutant constructs were generated using the Q5 site-directed mutagenesis kit (NEB). Primers used are indicated in Supplementary Table 2. Constructs used for BioID screen validation not already tagged with GFP, when possible were cloned into pEGFP vectors to more easily confirm expression and increased experimental consistency, details can be found in Supplementary Table 2. Transient transfection was performed using Lipofectamine and Plus reagents (Life Technologies) or Fugene-HD (Promega) according to the manufacturer's instructions, and assays were conducted 24–48 h post-transfection.

**Biochemical assays.** For all biochemical assays, unless otherwise stated cells were starved in HBSS 2 h prior to lysis to enhance autophagic flux. Experiments requiring Met-activation were done with stimulation with 0.5 nM HGF (generous gift from Genentech) for 20 min prior to lysis.

For co-immunoprecipitation studies, HeLa cells were transiently transfected with the appropriate constructs for 24 h. Cells were harvested in lysis buffer (50 mM HEPES, 150 mM NaCl, 1.5 mM MgCl₂, 1 mM EGTA, 1% Triton X-100, 10% glycerol, pH 7.4, 1 mM PMSF, 1 mM Na₃VO₄, 1 mM NaF, 10 μg/ml aprotinin and 10 μg/ml leupeptin). Lysates were pre-cleared with protein-A sepharose beads overnight at 4 °C. 1000 mg of protein was then incubated with 1.2 μl of Met-148

antibody (made in house), 1.5 µl anti-GFP (Invitrogen), 1.5 µl anti-Myc (Clonetech) or 1.5 µl anti-V5(Abcam) for 4 h at 4 °C followed by the addition of protein A-sepharose (GE Healthcare) for an additional 3 h of incubation. Beads with bound proteins were washed four times in lysis buffer plus inhibitors, and eluted by boiling in SDS sample buffer. Eluted proteins and 30 µg of protein from whole cell lysate were used for western blotting.

For analysis Met-RTK stability and TfR levels, cells were transiently transfected with the appropriate constructs for 24 h. To differentiate from any effects of synthesis in experiments studying degradation cells were treated with 100 µg/ml cycloheximide (Sigma) to inhibit translation for 1.5–2 h prior to lysis. For analysis of Met stability following the cycloheximide treatment, cells were stimulation with 0.5 nM HGF for the indicated time point prior to lysis.

**SDS–PAGE and western blotting**. Proteins were separated using 4–15% NuPage Gradient gels (Thermo Fisher Scientific) with the MES-SDS running buffer (Thermo Fisher Scientific). Proteins were transferred on PVDF Odyssey membranes (MilliporeSigma) using a Mini Trans-Blot System from Bio-Rad. Detection and quantification of protein levels were performed on the Odyssey IR imaging System (Li-COR Biosciences). All antibodies used in this study are listed in Supplementary Table 3, including information regarding dilutions.

**Immunofluorescent staining**. For all experiments, unless otherwise stated cells were starved in HBSS 2 h prior to fixation to enhance autophagic flux. For immunofluorescence experiments, cells were plated on coverslips (Bellco Glass Inc) 48 h prior to treatment and fixation. When necessary transfections were done 24 h after plating. Cells were fixed in 4% PFA (Fisher Scientific) for 20 min at room temperature. For staining cells were permeabilized in 0.3% Triton X-100 in PBS for 10 min. Blocking was in 2% BSA in washing buffer (PBS with 0.2% Triton X-100 and 0.05% Tween-20) for 30 min. Primary antibodies (1/75-1/100) and secondary antibodies (1/200-1/500) treatments were in blocking solution for 60 min and 45 min, respectively. Where applicable, nuclei were counterstained with 4′,6-diamidino-2-phenylindole (DAPI) (Sigma) prior to mounting using Immu-Mount (Thermo Shandon Inc). Confocal images were acquired on a Zeiss LSM800 laser scanning confocal microscope (Carl Zeiss) with a 63X objective and analyzed using Zen software (Carl Zeiss, version 2.3) and MetaMorph software (Molecular Devices, version 7.7.7.0)). All antibodies used in this study are listed in Supplementary Table 3.

**Met trafficking assays**. For experiments involving analysis of Met trafficking an immunofluorescence-based recycling assays were conducted. Cells were grown on glass coverslips 48 h prior to treatment and fixation. When necessary transfections were done 24 h after plating. To examine Met-trafficking cells were starved in HBSS with 100 µg/ml cycloheximide (Sigma) for 1.5 h, and cold loaded with 0.5 nM HGF in the presence of cycloheximide for another 45 min at 4 °C. Cells were washed once with prewarmed media and then treated for 6 min with pre-warmed media with HGF to induce internalization. HGF-media was removed by washing three times with 0.2% BSA in Leibovitz-15 medium at 4 °C, prior to a return to 37 °C for 30 min to allow for Met to recycle or proceed through the degradative pathway; at the end of which cells were fixed in 4% PFA (Fisher Scientific) for 20 min at room temperature and then stained.

**Cell permeabilization**. Cells were plated and treated as described for other immunofluorescence experiments. Following treatment media was aspirated and cells were permeabilized by the addition of permeabilization buffer consisting of ice cold 0.05% saponin in piperazine-N, N′-bis(2-ethanesulfonic acid) (PIPES) buffer (80 mM PIPES KOH pH 7.0, 5 mM EGTA, 1 mM MgCl$_2$) for 3 min.

**Live cell imaging**. Assays were performed 48 h after plating cells in an ibidi glass-bottom dish (81158), with transfections being done 24 h following plating. Media was aspirated, and cells were rinsed once with PBS and starved with HBSS for 1 h prior to image collection. For HGF-555 colocalization studies a bolus of HGF-555 was added to each plate (C$_f$ = 0.5 nM). Four different regions were imaged per plate every 5 s for ~10 min each, in the period 20–60 min post-stimulation. Images were captured with a TIRF-Spinning Disk Spectral Diskovery System (Spectral Applied Research, Richmond Hill, ON) based on a Leica DMI 6000 microscope stand (Quorum Technologies, Puslinch, ON) equipped with a Leica Plan-Apochromat 63x/1.47NA oil DIC objective, ImagEM X2 EM-CCD camera (Hamamatsu Photonics K.K., Hamamatsu City, Japan), and Chamlide CU-501 top-stage incubator system (Live Cell Instrument, Seoul, South Korea) using Meta-Morph software (Molecular Devices, version 7.7.7.0) acquisition and assembly. For cell permeabilization experiments, permeabilization buffer was added and a cluster of cells was imaged every 5 s for ~2–3 min.

**Flow cytometry**. Prior to staining HeLa cells were treated in the same manner as for the immunofluorescence trafficking experiments described above. Briefly, cells were starved in HBSS with 100 µg/ml cycloheximide (Sigma) for 2 h to differentiate from any effects of synthesis prior to fixation and staining. For analysis of Met surface levels, cells were in addition cold-loaded with 0.5 nM HGF at 4 ºC for 45 min, washed and incubated with prewarmed media for 6 min to induce Met-internalization and following HGF-washout returned to 37 °C for 30 min to allow for Met to recycle or proceed through the degradative pathway. For staining the following fluorescent-labeled antibodies were used: PE anti-MET (clone 95106; #FAB3582P R&D systems), APC anti-CD71 (clone CY1G4; #334108 Biolegend), PE mouse IgG1 κ isotype control (clone MOPC-21; #400114 Biolegend) and APC mouse IgG2a κ isotype control (clone MOPC-173; #400219 Biolegend). A fixable live/dead dye was used to distinguish viable cells (#423113 Biolegend). Cell surface staining was performed in FACS buffer (PBS supplemented with 0.5% BSA and 2 mM EDTA). Stained cells were acquired on a LSR Fortessa flow cytometer (BD Biosciences) at the Flow Cytometry Core Facility of the Life Science Complex supported by funding from the Canadian Foundation for Innovation. Data were analyzed with FlowJo LLC software (BD Biosciences, version X 10.0.7r2). All antibodies and dilutions used are listed in Supplementary Table 3.

**Quantification and statistical analysis**

*Colocalization analysis*. Colocalization analysis was performed using object-based colocalization quantification in MetaMorph software (Molecular Devices; version 7.7.7.0). Using the 'granularity' application, puncta were identified using an intensity and size threshold and a binary mask was created for each channel individually. The masks of each channel were multiplied so that only pixels present in both images were retained. Puncta in all binary images were then counted using the 'count object' function. All colocalization quantification was performed on a minimum of 40 cells (minimum 10 cells per experimental replicate).

*Puncta distribution analysis*. Puncta distribution analysis was performed using MetaMorph software (Molecular Devices; version 7.7.7.0). Individual cells were traced manually. The nucleus was detected in the DAPI-channel, and the PM was detected using either phalloidin or another stain with enough cytoplasmic presence to delineate the cell limits. A distance map was created where values indicate the Euclidian distance of any point to either the nucleus-mask or the PM-mask. Using the 'granularity' application, puncta were identified using an intensity and size threshold and a binary mask of the puncta was created. The puncta-mask was then transferred separately to the 'Nucleus-Euclidian Distance Map' and 'PM-Euclidian Distance Map' so that the distance of each individual puncta to the nucleus (d$_{nuc}$) and PM (d$_{PM}$) could be obtained. To account for irregular cell shapes the relative distance (d$_{rel}$) to the plasma membrane was finally calculated: d$_{rel}$ = d$_{PM}$/(d$_{PM}$ + d$_{nuc}$). For distribution analysis, each cell was divided into three cellular regions defined as Peripheral (0 < d$_{rel}$ ≤ 0.3), Intermediate (0.3 < d$_{rel}$ ≤ 0.7) and Peri-nuclear (0.7 < d$_{rel}$ ≤ 1), and the proportion of puncta in each region was calculated using a custom R script. For each condition a minimum of 50 cells were used from samples collected across a minimum of three experimental replicates.

*Live cell movie analysis*. Live cell movie analysis was performed using Cell Profiler software (version 4.0.4)[69]. For individual cells, ATG13 or FIP200 puncta were detected in the GFP channel using an intensity and size threshold (with a shape smoothing function), these puncta were tracked across the length of the movie using the 'Object Tracking' function. The GFP-objects were overlayed to the mCherry channel, and the average red fluorescence was measured for each object per frame. The threshold of what constitutes a positive HGF-555 puncta was measured. GFP-objects that displayed average red fluorescence above this threshold for a minimum of 15 s were counted as becoming positive for HGF-555 during the imaging period. A minimum of 20 cells were quantified per condition.

**Statistics**. Data were analyzed using R programming environment (v.3.6.2, r-project.org via R-Studio interface v.1.2.5033, rstudio.com) and visualized using the ggplot2 package (v.3.3.3, r-project.org). Quantitative data are presented as the means ± standard error of the mean (SEM). Statistical significance was assessed using unpaired, two-tailed Student's t-test. p-values and the number of independent experiments (N) used for quantification and statistical analysis are indicated in the corresponding figures and figure legends. For normalized data, a two-tailed one sample t-test was used to determine significant difference from a hypothetical value of 1.0. For enrichment analysis of the manually curated list of protein location a Fischer-test was used. For enrichment analysis of protein location done using the Subcellular Barcode database a Hypergeometric test was used. Values were considered to be significant when p values were <0.05.

**Reporting summary**. Further information on research design is available in the Nature Research Reporting Summary linked to this article.

## Data availability

The data that support this study are available from the corresponding author upon reasonable request. The mass spectrometry data generated in this study have been deposited in the MassIVE (http://massive.ucsd.edu) database under accession code MSV000087983. The ProteomeXchange accession is PXD027926. As indicated above, subcellular localization analysis was performed using data from the SubCellBarcode database (subcellbarcode.org). The increased coverage of subcellular location

classification was achieved with additional Gene Ontology (GO) data downloaded from BioMart (Esembl Gene 88; useast.ensembl.org/info/data/biomart/index.html), supplemented with information from GeneCards (genecards.org) and COMPARTMENTS database (compartments.jensenlab.org). Source data are provided with this paper.

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

## Acknowledgements

We thank members of the Park laboratory for their helpful comments on the manuscript; and Genentech Inc. for HGF. Images were collected for this manuscript was performed in the McGill University Advanced BioImaging Facility (ABIF), RRID:SCR_017697. Proteomics work was performed at the Network Biology Collaborative Centre at the Lunenfeld-Tanenbaum Research Institute, a facility supported by Canada Foundation for Innovation funding, by the Ontario Government, and by Genome Canada and Ontario Genomics (OGI-139). The flow cytometry work was performed in the Flow Cytometry Core Facility of the Life Science Complex and supported by funding from the Canadian Foundation for Innovation. This research was supported by doctoral studentships from the Fonds de Recherche du Québec – Santé to P.P.C. and C.D.H.R. and the Rosalind Goodman Commemorative Scholarship to P.P.C. and C.D.H.R.; the United States Department of Defense (BC100836) to E.S.B.; the GCRC Recruitment Scholarship to A.P.; and Foundation operating grants to N.So. (148423), M.P. (242529) and A.C.G. (143301) from the Canadian Institutes of Health Research. M.P. holds the Diane and Sal Guerrera Chair in Cancer Genetics. And A.C.G. is the Canadian Research Chair in Functional Proteomics.

## Author contributions

Conceptualization, P.P.C. and M.P.; Methodology, P.P.C., G.G.H., E.K., and C.D.H.R.; Formal analysis, P.P.C., G.G.H., and E.K.; Investigation, P.P.C., G.G.H., A.P., A.M.N.F., and E.S.B.; Resources A.C.G. and M.P.; Writing—original draft P.P.C. and M.P.; Writing—review and editing: all authors; Supervision: A.C.G. and M.P.

## Competing interests

The authors declare no competing interests.
