## [Peer Review File · Nature Communications]

Endosomal LC3C-pathway selectively targets plasma membrane cargo for autophagic degradationReviewers' Comments:

Reviewer #1:

Remarks to the Author:

Coelho et al. focus on the biological functions of LC3C, which remains the most enigmatic among the ATG8 isoforms. To better understand the functions of LC3C, the authors perform a proximity-based proteomics screen comparing the interactors of LC3B versus LC3C in HELA cells. This leads to the discovery that LC3C is involved in the transfer of proteins that are internalized via the early endocytic trafficking pathway, namely HGF/Met, into the classical autophagy pathway via the interaction of LC3C with ATG9 vesicles. This pathway, which is termed LEAP, is dependent on TBK1 phosphorylation of specific sites on LC3C as well as on the C-terminal tail.

Overall, the results are interesting but the conclusions on LC3C are overly reliant on co-location and co-immunoprecipitation experiments rather than genetic interrogation of key players. Importantly, no detailed analysis on the functional role of core autophagy machinery in the control of LEAP is conducted, which is a major conceptual gap. Hence, it is unclear from this model with regard to the requirement for localizing LC3C to the early endocytic compartments as well as ascertaining whether LEAP represents a LAP-like autophagy-related pathway. Additional studies are necessary to more rigorously support the conclusions of the paper.

1) From the studies and the model proposed, it appears that LC3C conjugation may be occurring onto a single membrane at the early endosome which would be akin to other recently described pathways such as LAP, LANDO and LDELS. However, genetic support for such a model is lacking, which is the major weakness of the paper. Is the ability of LC3C to localize to peripheral endosome specifically dependent on the LC3 conjugation pathway? Studies comparing the effects of ATG7, ATG3 versus FIP200, ATG14 loss-of-function should provide the necessary insight into this critical issue.

2) Is the proximity proteomics analysis of LC3B vs LC3C conducted during starvation conditions or is this all performed in the context of basal autophagy. It seems that certain studies in the paper utilize starvation to induce autophagy whereas others do not, which makes it difficult to ascertain whether LEAP is a constitutive trafficking process or one that functions upon stimulation of classical starvation-induced autophagy.

3) LC3 binding to TBC1D5 has been implicated and cell-surface trafficking of nutrient transporters, such as GLUT1, via control of the retromer (Roy et al. Molecular Cell, 2017 PMID: 28602638) although the role of specific LC3 isoforms were not dissected in that previous study. Is the retromer involved in the control of LEAP? Do LC3C or LC3B differentially bind to TBC1D5?

4) The authors propose that LC3C and LEAP controls the cell surface expression of TRFC and Met on cells. Formal evidence for this conclusion should be provided via flow cytometry for these two receptors.

Reviewer #2:

Remarks to the Author:

In this manuscript, the authors report a novel autophagic pathway for the degradation of plasma membrane cargoes by autophagy. This pathway depends on LC3C including its C-terminal tail stabilized by phosphorylation by TBK1, which is required for the incorporation of plasma membrane proteins into ATG9A and VAMP3 positive compartments. In principle, the concept of plasma membrane degradation by macroautophagy is interesting. However, as detailed below I feel that the study is too preliminary to make the claim that LC3C-dependent autophagy degrades the plasma membrane proteins studied here.

1. The title of the manuscript suggests that LC3C is required for the degradation of plasma membrane proteins by autophagy (given the factors studied in the manuscript, I presume the term autophagy refers to macroautophagy). In particular, the authors study the co-localization of the Met and TFRC receptors with autophagy proteins LC3C and ATG9A. However, no actual degradation is analysed in this manuscript. Is the (macro)autophagic degradation of these protein dependent on LC3C, its C-terminal tail and TBK1 phosphorylation? If so, at which step? How would this be mediated? For example, are small fragments of plasma membrane present in autophagosomes and the delivered into lysosomes independent of the ESCRT system? Alternatively, if the cargoes would be incorporated into the autophagosomal membrane for degradation, then this would have to be the inner membrane because this is the membrane delivered into lysosomes. According to the model in Figure 6E, the ligand binding domains of the receptors are inserted into both autophagosomal membranes while the transmembrane domain protrudes into the lumen.

2. Throughout the manuscript LC3B and LC3C are ectopically expressed, presumably resulting in much higher expression levels compared to the endogenous proteins, which can result in overexpression artifacts. The experiments should therefore be repeated, either by using validated antibodies against the endogenous proteins or by genomically tagging them. A similar concern applies to VAMP3, which is also overexpressed. As a transmembrane protein it may spill over to other compartments compromising co-localization studies.

3. The authors use BirA* fused LC3B and LC3C proteins to identify proteins, which come into close proximity to them. Since the stringency of this approach is very much dependent on the expression levels of the baits, their expression levels relative to the respective endogenous proteins should be shown and if necessary adjusted.

Reviewer #3:

Remarks to the Author:

Paula P. Coelho et al describe the investigation of the proximal interactome of LC3B and LC3C, obtained using BioID. They show specific interactors for both LC3B and LC3C, where the latter localize to peripheral endosomes and bind cargo internalized from the plasma membrane (e.g. MET). Next, they show the unique features of LC3C are related to its C-term tail and show that binding and phosphorylation of the tail region regulate LC3C localization and cargo interaction. They term this new pathway LEAP. The study is well executed and clearly described, the BioID data seems robust and findings are well validated. There are only some minor comments.

Specific comments.

The described phosphorylation of LC3C by TBK1 is based on literature and observed phenotype of the mutant strain, which behaves like the TBKi condition. The actual phosphosite and its loss upon TBK inhibition has not been shown. Did the authors search the BioID data for the presence of phosphosites or try a phosphopeptide enrichment to validate it's actually the ser93/96 phosphorylation that sequesters the C-term tail?

Suppl figure 5 shows the sequence of LC3C and specifically the p-sites and C-tail. The cartoon shows how phosphorylation of ser93/96 sequesters the C-tail but there is no mention of how this conformation make sense in relation to the negative charge of the p-sites and the sequence of the C-tail. Can the authors elaborate on this?

Reviewer #1 (Remarks to the Author):

Coelho et al. focus on the biological functions of LC3C, which remains the most enigmatic among the ATG8 isoforms. To better understand the functions of LC3C, the authors perform a proximity-based proteomics screen comparing the interactors of LC3B versus LC3C in HELA cells. This leads to the discovery that LC3C is involved in the transfer of proteins that are internalized via the early endocytic trafficking pathway, namely HGF/Met, into the classical autophagy pathway via the interaction of LC3C with ATG9 vesicles. This pathway, which is termed LEAP, is dependent on TBK1 phosphorylation of specific sites on LC3C as well as on the C-terminal tail.

1) From the studies and the model proposed, it appears that LC3C conjugation may be occurring onto a single membrane at the early endosome which would be akin to other recently described pathways such as LAP, LANDO and LDELS. However, genetic support for such a model is lacking, which is the major weakness of the paper. Is the ability of LC3C to localize to peripheral endosome specifically dependent on the LC3 conjugation pathway? Studies comparing the effects of ATG7, ATG3 versus FIP200, ATG14 loss-of-function should provide the necessary insight into this critical issue.

We appreciate the reviewer's comments and have conducted additional experiments to assess dependency on conjugation for LC3C LEAP interactions and LC3C localization. We show that following knock-down (KD) of components of the conjugation machinery (ATG3, ATG5 and ATG7), LC3C retains ability to coimmunoprecipitate with cargo, Met and TFRC (new Fig. 6a, b). Hence LC3C interaction with cargo is independent of the conjugation machinery. In support of this, following KD of core conjugation machinery, ATG7, we still observed peripheral vesicle-bound LC3C puncta after permeabilizing cells to remove unbound cytosolic LC3C, which results from decreased LC3C conjugation to autophagic membranes in ATG7-KD cells (new Supplementary Fig. 6c and new Supplementary Mov. 1). Notably these remaining LC3C puncta colocalize with Met following HGF stimulation as well as endocytic markers Rab4 and Rab5 (new Fig. 6c, d). Together this new data provides strong evidence supporting engagement of LC3C with Met and TFRC prior to LC3C cleavage and lipidation.

As requested, we also assessed the role of components of the autophagy initiation and nucleation machinery (FIP200 and ATG14 respectively). Following KD of FIP200, transfection of LC3C led to reproducibly lower levels of LC3C making analysis of LC3C interactions and localization under these conditions inappropriate and may reflect additional functions for FIP200 (data provided for the reviewer). In contrast KD of either ATG14 or ATG13, another member of the initiation complex, did not detectably alter LC3C protein levels, and so these were used for further experiments. We found that following KD of ATG14 or ATG13, LC3C still coimmunoprecipitates with cargo (Met and TFRC; new Supplementary Fig. 6a, b). Moreover, as shown for KD-ATG7, following cell permeabilization in ATG14-KD cells, LC3C colocalized with Met and endocytic markers Rab4 and Rab5 (new Supplementary Fig. 6c, d). Overall, these new data (Fig. 6 and Supplementary Fig. 6), provide strong evidence supporting a role for LC3C prior to its conjugation to autophagosomes, whereby unlipidated LC3C can engage with plasma membrane proteins, Met and TFRC, in a peripheral endosomal compartment.

2) Is the proximity proteomics analysis of LC3B vs LC3C conducted during starvation conditions or is this all performed in the context of basal autophagy. It seems that certain studies in the paper utilize starvation to induce autophagy whereas others do not, which makes it difficult to ascertain whether LEAP is a constitutive trafficking process or one that functions upon stimulation of classical starvation-induced autophagy.

*We agree with the reviewer that clarification on the role of starvation in LEAP would be beneficial. The BioID screen was performed under conditions of basal autophagy. Previous proteomic interaction studies into the autophagic machinery found that increased autophagy (due to Torin-dependent mTOR inhibition) did not lead to large-scale changes in core conjugation, lipid kinase and recycling complexes (Behrends et al. Nature, 2020. PMID: 20562859). The authors of that study noted however that interaction of ATG8-protens with some known cargo receptors was reduced upon Torin-dependent autophagy induction, which they postulate was caused by the degradation of such cargo-receptors upon autophagic delivery to the lysosome. Thus, we postulated that performing the BioID screen under basal conditions was most appropriate to provide a readout of LC3-interactomes, without risk of losing interactors via increased degradation. As the reviewer notes however, we did seek to expand and validate the discoveries made in our BioID screen under conditions of a starvation-stress, thereby also providing insight into the role of LEAP under conditions where the autophagic flux is enhanced. Thus, the BioID screen results combined with multiple experimental approaches, indicate that LEAP is likely a constitutive trafficking process under basal autophagy that is enhanced upon starvation as the autophagic flux increases. We agree with the reviewer however that directly addressing this point would be of value to the manuscript and have added experiments showing that LC3B and LC3C have distinct distribution under basal autophagy conditions (**new Supplemental Fig. 2b**), which are enhanced upon starvation (Fig. 2a, b). We have also clarified the BioID screen conditions used in our study in the manuscript.*

3) LC3 binding to TBC1D5 has been implicated and cell-surface trafficking of nutrient transporters, such as GLUT1, via control of the retromer (Roy et al. Molecular Cell, 2017 PMID: 28602638) although the role of specific LC3 isoforms were not dissected in that previous study. Is the retromer involved in the control of LEAP? Do LC3C or LC3B differentially bind to TBC1D5?

The reviewer brings up an interesting point regarding the regulation of TBC1D5 by autophagy, and consequent changes in GLUT1 trafficking via the retromer. Although the study cited does largely dissect the regulation of TBC1D5 by autophagy without great emphasis on LC3-orthologs, the authors did address the specificity of LC3 proteins and claim that “TBC1D5 strongly interacts with LC3A and to a lesser extent with LC3C”, in data provided in the supplemental material.

Whereas our study did not interrogate LC3A, we detected TBC1D5 as a high-confidence LC3B proximity-interactor (FDR < 0.01), but not as a high confidence LC3C proximity-interactor (it was detected amongst the LC3C-proximal interactome but with an FDR = 0.09). Since LEAP relies on ligand induced internalization of the Met RTK (blocked by dynasore) we have no evidence to implicate the retromer in the control of LEAP.

4) The authors propose that LC3C and LEAP controls the cell surface expression of TRFC and Met on cells. Formal evidence for this conclusion should be provided via flow cytometry for these two receptors.

We thank the reviewer for this suggestion. We have performed additional experiments quantifying surface levels of Met and TFRC by flow cytometry. Consistent with our data from immunofluorescence-based assays, LC3C deletion leads to higher cell surface levels of Met following HGF-stimulation and of TFRC (new Fig. 4d,k and Supplementary Fig. 4a,f).

Reviewer #2 (Remarks to the Author):

In this manuscript, the authors report a novel autophagic pathway for the degradation of plasma membrane cargoes by autophagy. This pathway depends on LC3C including its C-terminal tail stabilized by phosphorylation by TBK1, which is required for the incorporation of plasma membrane proteins into ATG9A and VAMP3 positive compartments. In principle, the concept of plasma membrane degradation by macroautophagy is interesting.

1. The title of the manuscript suggests that LC3C is required for the degradation of plasma membrane proteins by autophagy (given the factors studied in the manuscript, I presume the term autophagy refers to macroautophagy). In particular, the authors study the co-localization of the Met and TFRC receptors with autophagy proteins LC3C and ATG9A. However, no actual degradation is analysed in this manuscript. Is the (macro)autophagic degradation of these protein dependent on LC3C, its C-terminal tail and TBK1 phosphorylation?

We thank the reviewer for this comment. We do refer to macroautophagy in the manuscript, as indicated in the introduction. We agree that additional experiments measuring the degradation of Met and TFRC receptors would be beneficial. We have performed extensive work on the LC3C-dependent autophagic degradation for the Met-receptor in a previous publication (Bell et al. Cell Reports, 2019, PMID: 31851933) and have referenced this more explicitly in the manuscript. We also performed additional new experiments measuring the LC3C-dependent degradation of TFRC, in addition to Met and have assessed the role of the LC3C-C-tail and LC3C TBK1-phosphosites in degradation of both receptors (new Supplementary Fig. 5c,d). We show that LC3C deletion led to delayed degradation of Met and of TFRC under starvation conditions, and that overexpression of LC3C-WT, but not LC3C-ΔCtail or LC3C-S93/96A, could rescue these phenotypes. These data support that both the LC3C C-terminal tail and LC3C TBK1-dependent phosphosites, beyond regulating LC3C localization, are also necessary for degradation of LC3C cargo, Met and TFRC.

If so, at which step? How would this be mediated? For example, are small fragments of plasma membrane present in autophagosomes and then delivered into lysosomes independent of the ESCRT system? Alternatively, if the cargoes would be incorporated into the autophagosomal

membrane for degradation, then this would have to be the inner membrane because this is the membrane delivered into lysosomes. According to the model in Figure 6E, the ligand binding domains of the receptors are inserted into both autophagosomal membranes while the transmembrane domain protrudes into the lumen.

We agree with the reviewer that these are important questions that would require a careful and detailed study to provide clarity. At this stage we feel that this is outside the scope of the current manuscript. We agree that there is evidence from multiple studies showing that PM-derived vesicles as well as endosomes can donate membrane to nascent phagophores (Sørensen et al. International Review of Cell and Molecular Biology, 2018, PMID: 29413888), making these well accepted membrane sources for autophagosomes allowing for possible incorporation of PM derived cargo into autophagosomal membranes. In addition, several previous studies have also shown that autophagy contributes to the degradation of membrane-receptors without mechanistic understanding (Sandilands et al. EMBO Rep, 2012, PMID: 22732841, Hsueh et al. Oncotarget, 2014, PMID: 25375091; Winer et al. Nat Commun, 2018, PMID: 30218067; Rao et al, Nat Commun, 2020, PMID: 32385243). We agree with the reviewer that without direct evidence of PM cargo in the autophagic-membrane itself this remains a model and we have stressed this in the text. All of our data support that Met does arrive at preautophagosomal structures (PAS) with ATG9 in an LC3C-dependent manner. Not only did we establish Met colocalizes with ATG9, but we also show that a Met-HGF complex colocalizes with both GFP-ATG13 and GFP-FIP200, markers of nascent autophagosomes. This supports that Met is delivered to PAS where it can then be incorporated into autophagosomes. We have now updated the model in Fig. 7e to depict the targeting of the receptors to autophagosomes in a more general manner.

2. Throughout the manuscript LC3B and LC3C are ectopically expressed, presumably resulting in much higher expression levels compared to the endogenous proteins, which can result in overexpression artifacts. The experiments should therefore be repeated, either by using validated antibodies against the endogenous proteins or by genomically tagging them.

We appreciate the limitation of ectopically expressing LC3B and LC3C, however the LC3C-specific antibodies we have tested extensively have also recognized other LC3 proteins and have a non-specific staining pattern by immunofluorescence, limiting their use. We have thus aimed to supplement our findings by using isogenic LC3C-deletion cells in part to avoid the effects of overexpression in these experiments. For example, by assessing endogenous Met and TFRC trafficking and colocalization with ATG9 in isogenic control and LC3C-deletion cell lines (Fig. 4a-c and Fig. 4h-j) and cell-surface expression and protein levels of endogenous Met and TFRC (new data in Fig. 4d,k and Supplementary Fig. 5c,d). These experiments add additional conditions addressing the role of LC3C and LEAP without possible artifacts that may result from overexpression. We also note that the system we have selected for the proteomics screen (the Flp-In T-REx system) enables for the selection of cells with a single integration of the tagged transgene, under the control of an inducible promoter, which in our experience, does not lead to the type of massive overexpression seen in typical transient transfection experiments (St-Denis, et al. Cell Reports, 2016, PMID: 27880917). While we acknowledge that genomically tagging these constructs would be of value, by combining overexpression and deletion studies we are able to interrogate the role of LC3C and derive valid and applicable conclusions of its role in cells. Moreover, as controls, we had compared how varying levels of LC3B and LC3C expression impact

their distribution within cells and established that 3-4 fold variation in LC3B or LC3C protein expression did not lead to significant changes in the distribution of LC3B or LC3C puncta, indicating this is not an artifact of overexpression. We have included the data below for your interest and could add to a supplemental figure if requested.

Legend: *Top left:* western blot analysis of expression levels of Myc-LC3B and Myc-LC3C following transient transfection. *Top right:* quantification of blots on the left, display fold change in normalized Myc-LC3 levels (Myc/Tubulin) from lowest expression condition. *Bottom:* Quantification of subcellular distribution of Myc-LC3 puncta in of cells expressing varying amounts of Myc-LC3; the fraction of puncta identified across three subcellular cellular regions (peripheral, intermediate and perinuclear) are plotted.

A similar concern applies to VAMP3, which is also overexpressed. As a transmembrane protein it may spill over to other compartments compromising co-localization studies.

We thank the reviewer for this comment and agree that it would be ideal to image endogenous VAMP3. However, the antibody we were able to validate by western-blot was not recommended for use by immunofluorescence, and in fact displayed a non-specific staining pattern when tested, with many small puncta detected in shVAMP3 cells, where almost no protein was observed by immunoblotting. This prevented its use in the colocalization studies. Moreover, our overexpression experiments were performed with approximately 2X the endogenous levels of VAMP3 as established by immunoblotting, and therefore did not significantly alter the amount of VAMP3 in the cells imaged, which we believe allows for accurate interpretation of the colocalization experiments. We have included this data below for the reviewer.

Legend: Top right: western-blot analysis of VAMP3 expression following VAMP3-KD with two different guides; Top left: immunofluorescence staining of empty vector control (pLKO.1) and shVAMP3#2 stained with endogenous VAMP3 antibody; Bottom: western-blot analysis of HeLa cells transiently transfected with varying amounts of GFP-VAMP3 with the quantification of blots on the right, GFP-VAMP3 levels were normalized to the corresponding endogenous VAMP3.

3. The authors use BirA* fused LC3B and LC3C proteins to identify proteins, which come into close proximity to them. Since the stringency of this approach is very much dependent on the expression levels of the baits, their expression levels relative to the respective endogenous proteins should be shown and if necessary adjusted.

We agree with the reviewer that BioID methodology is designed to identify proximity-interactors rather than direct interactors and have referred to our screen results as such. Many proteomic screens do rely on overexpression methods and we acknowledge the limitations that might arise from this. Consequently, we used a used a BirA-GFP and BirA*-FLAG alone as controls, which is helpful to model the background due to expression of high levels of an active biotin ligase enzyme. We then used a stringent Significance Analysis of INteractome cut-off FDR<0.01(1%) for scoring to decrease false-positives, and proceeded to validate interactions with many hits, including all the proteins we propose to play a role in LEAP. Moreover, we supplement our findings by genetic manipulation of LC3C to further support our conclusion in a non-overexpressed setting. Thus, we have provided multiple experimental support of the validity and usefulness of the screen as a tool to uncover new roles for these LC3-proteins. We have added additional data showing the levels of BirA* fused LC3B and LC3C, demonstrating that these proteins are expressed at similar levels (new Supplementary Fig. 1a).*

Reviewer #3 (Remarks to the Author):

Paula P. Coelho et al describe the investigation of the proximal interactome of LC3B and LC3C, obtained using BioID. They show specific interactors for both LC3B and LC3C, where the latter

localize to peripheral endosomes and bind cargo internalized from the plasma membrane (e.g. MET). Next, they show the unique features of LC3C are related to its C-term tail and show that binding and phosphorylation of the tail region regulate LC3C localization and cargo interaction. They term this new pathway LEAP. The study is well executed and clearly described, the BioID data seems robust and findings are well validated. There are only some minor comments.

Specific comments.

The described phosphorylation of LC3C by TBK1 is based on literature and observed phenotype of the mutant strain, which behaves like the TBKi condition. The actual phosphosite and its loss upon TBK inhibition has not been shown. Did the authors search the BioID data for the presence of phosphosites or try a phosphopeptide enrichment to validate it's actually the ser93/96 phosphorylation that sequesters the C-term tail?

Suppl figure 5 shows the sequence of LC3C and specifically the p-sites and C-tail. The cartoon shows how phosphorylation of ser93/96 sequesters the C-tail but there is no mention of how this conformation make sense in relation to the negative charge of the p-sites and the sequence of the C-tail. Can the authors elaborate on this?

*We thank the reviewer for this comment, and although our BioID data provides us with extensive data regarding LC3-proximal interactors, the relatively low protein coverage as well as the lack of phosphopeptide enrichment step in our protocol do not allow us to extrapolate the status of phosphosites on the LC3-proteins themselves. Given that, we detected TBK1 as an LC3C-specific proximal interactor, and in the context of the detailed work by Herhaus et al. (EMBO-reports, 2020, PMID: 31709703) that characterized the TBK1-dependent LC3C-phosphorylation, we elected to study the function of TBK1 in the regulation of LEAP. Herhaus et al (EMBO-reports, 2020) identified the S93-96 as TBK1-dependent phosphorylation sites on LC3C following mass spectrometry experiments and provide extensive information on how the phosphorylation of Ser93/96 acts to stabilize the LC3C-Ctail. Specifically, the authors of that study showed that phospho-S93/96 could form an intramolecular salt bridge with R134 on LC3C, acting to pull the C-terminal tail of LC3C toward the protein. We acknowledge it would be beneficial for the readers to have this information and have included this in the manuscript (pg 11), as well as in the model used in **Supplementary Fig. 5a**. Thus, although we did not perform additional mass spectrometry experiments to further analyze the status of LC3C-phosphorylation, we believe that our structure function data showing a requirement for Ser 93/96 for localization of LC3C to peripheral puncta (Fig. 5b), and degradation of Met and TFRC (**new Supplementary Fig. 5c,d**); together with a requirement for TBK1 activity for localization of LC3C to peripheral puncta (Fig. 7b) and for coimmunoprecipitation with Met or TFRC (Fig. 7d), provide strong experimental support for the involvement of these serine-residues and TBK1 in the regulation of LEAP.*

Reviewers' Comments:

Reviewer #1:

Remarks to the Author:

I have read over the revised version and author's response. My previous concerns have been satisfied.

Reviewer #2:

Remarks to the Author:

The authors have answered my comments and I have no further points.